# Generation of mesenchyme free intestinal organoids from human induced pluripotent stem cells

Aditya Mithal [1,2], Amalia Capilla [1], Dar Heinze [1,3], Andrew Berical[1,4], Carlos Villacorta-Martin[1], Marall Vedaie[1], Anjali Jacob[1], Kristine Abo [1], Aleksander Szymaniak[5], Megan Peasley[5], Alexander Stuffer[5], John Mahoney[5], Darrell N. Kotton[1,4], Finn Hawkins [1,4] & Gustavo Mostoslavsky [1,2,6]*

Efficient generation of human induced pluripotent stem cell (hiPSC)-derived human intestinal organoids (HIOs) would facilitate the development of in vitro models for a variety of diseases that affect the gastrointestinal tract, such as inflammatory bowel disease or Cystic Fibrosis. Here, we report a directed differentiation protocol for the generation of mesenchyme-free HIOs that can be primed towards more colonic or proximal intestinal lineages in serum-free defined conditions. Using a *CDX2eGFP* iPSC knock-in reporter line to track the emergence of hindgut progenitors, we follow the kinetics of *CDX2* expression throughout directed differentiation, enabling the purification of intestinal progenitors and robust generation of mesenchyme-free organoids expressing characteristic markers of small intestinal or colonic epithelium. We employ HIOs generated in this way to measure *CFTR* function using cystic fibrosis patient-derived iPSC lines before and after correction of the *CFTR* mutation, demonstrating their future potential for disease modeling and therapeutic screening applications.

[1] Center for Regenerative Medicine of Boston University and Boston Medical Center, 670 Albany Street, Boston, MA 02118, USA. [2] The Department of Microbiology at Boston University School of Medicine, 700 Albany Street, Boston, MA 02118, USA. [3] The Department of Surgery at Boston University School of Medicine, 72 E Concord Street, Boston, MA 02118, USA. [4] The Pulmonary Center at Boston University School of Medicine, 72 E Concord Street, Boston, MA 02118, USA. [5] Cystic Fibrosis Foundation Therapeutics Lab, 44 Hartwell Avenue, Lexington, MA 02421, USA. [6] The Section of Gastroenterology in the Department of Medicine at Boston University School of Medicine, 650 Albany Street, Boston, MA 02118, USA. *email: gmostosl@bu.edu

Three-dimensional tissue-specific organoids represent a powerful tool to study both normal development and disease. Organoids have been generated from a variety of primary tissue samples, including small intestine[1,2], stomach[3], colon[4], and pancreas[5]. Since the discovery of the Wnt-activated $LGR5^+$ stem cell niche at the base of small intestinal and colonic crypts[1], previous studies have reported the generation of 3D intestinal organoids containing crypt-like structures from murine and human $LGR5^+$ intestinal stem cells in the presence of Wnt stimulation, epidermal growth factor (EGF) signaling, and Noggin[2]. However, the invasive procedures to obtain intestinal and colonic biopsy samples present a major challenge for larger scale applications of human intestinal organoids. The discovery of induced pluripotent stem cells (iPSCs)[6] has led to the development of multiple directed differentiation protocols, resulting in the in vitro generation of various endoderm-derived tissue types of interest, including liver[7], stomach[8], pancreas[9], proximal[10–12] and distal[13] lung, kidney[14], as well as intestine[15]. Moreover, the three-dimensional culture systems that generate organoids allow cells to self-organize, promoting further maturation and differentiation into target cell types that more closely resemble their in vivo counterparts[16,17].

The efficient generation of iPSC-derived human intestinal organoids (HIOs) serves not only as a relevant tool to study development, but has great potential for patient-specific in vitro disease modeling and high-throughput drug screening applications. HIOs positive for intestinal markers such as the intestinal homeobox transcription factor Cdx2[18,19] and intestinal epithelium marker Cdh17 have been generated from iPSCs using activin A to derive $SOX17^+/FOXA2^+$ endoderm, followed by Wnt3A and FGF4 (with serum) to specify $CDX2^+$ hindgut (Hindgut Medium), and R-spondin, EGF, and the BMP inhibitor, noggin (Intestinal Medium or IM) to promote intestinal specification and crypt-like formation[15]. More recently, distal patterning of iPSC-derived HIOs to generate $SATB2^+$ colonic organoids was achieved through BMP2 stimulation[20]. These factors have all been shown to play a role in intestinal specification and epithelial proliferation during embryonic development[21]. Interestingly, this protocol often generates HIOs containing both epithelial and mesenchymal stromal cells[15,20], necessitating a FACS-based approach to isolate epithelial cell adhesion molecule positive (EpCAM$^+$) cells in order to interrogate epithelial-specific populations[22], complicating their use in disease modeling or drug screening applications to isolate epithelial-specific factors. The derivation of HIOs from intestinal crypts using the $LGR5^+$ adult stem cell population can generate organoids in the absence of mesenchyme[2], raising questions as to whether intestinal progenitors derived from iPSCs are comparable to native crypts in generating HIOs. Moreover, a directed differentiation protocol using fully defined culture conditions is still lacking, as current protocols rely on the addition of exogenous serum.

Here we describe a protocol using a well-defined, serum-free media for the robust de novo generation of epithelial iPSC-derived HIOs devoid of mesenchyme. In addition, we report the generation of a hiPSC CDX2-GFP reporter line that highlights the role of $CDX2$ as a specific marker for the emergence of iPSC-derived intestinal progenitors. This platform enables the study of both normal development as well as disease states of the gut (exemplified by cystic fibrosis), supporting the generation of patient-specific iPSC-derived organoids for interrogation, genetic manipulation, and large-scale drug screening applications.

## Results

**Generation of intestinal progenitors from iPSCs.** We and others have previously shown that dual-smad inhibition of the BMP/TGFβ signaling pathways (with dorsomorphin and SB431542) in definitive endoderm derived from iPSCs and ESCs promotes the development of endoderm competent to form anterior foregut derivatives, such as $NKX2$-1 positive lung or thyroid lineages[10–13,23,24]. Indeed, we performed fluorescence activated cell sorting (FACS) of cells expressing the anterior foregut endodermal transcription factor $NKX2$-1 or a combination of cell surface markers CD47$^{hi}$/CD26$^{lo}$ (NKX2-1$^+$) to enrich for a population of progenitors which can then be differentiated into proximal and distal lung lineages from human iPSCs[11–13]. In this protocol, prior single-cell sequencing of day 15 progenitors revealed the presence of cells expressing non-lung endodermal markers, including CDX2, and these non-lung lineages were enriched in the $NKX2$-1 negative fraction of cells (refs. [25,26] and Supplementary Fig. 1). Thus, we sought to investigate the potential of this differentiation approach to obtain intestinal organoids in defined, mesenchyme-free (MF) and serum-free culture conditions, in comparison to the previously described mesenchyme-containing (MC) protocol[15] (Fig. 1a).

Two independent human iPSC lines, bBU1c2[27] and BU3-$NKX2$-1$^{GFP}$-$SFTPC^{tdTomato}$ [11,13] (BU3NGST) were differentiated into CXCR4/c-Kit$^{+/+}$ definitive endoderm, then treated with dual-smad inhibition as described above. Endodermal cells were then further incubated in conditions to promote lineage specification through Wnt activation with the GSK3β inhibitor CHIR99021 (CHIR), BMP4, and retinoic acid (RA) (ref. [13] and Fig. 1a). At day 15, cells were sorted to isolate the $NKX2$-1$^+$ and $NKX2$-1$^-$ fractions using a published cell sorting algorithm developed by our group[11], based on either CD47$^{hi/lo}$ (for BU1) or $NKX2$-1$^{GFP+/-}$ (for BU3NGST) and plated into 3D Matrigel droplets. Both sorted populations were cultured in a defined, serum-free media containing CHIR and KGF together with Dexamethasone, cAMP and IBMX (CK + DCI), previously shown by our group to generate type II alveolar epithelial cells from $NKX2$-1$^+$ lung progenitors[13]. Both iPSC lineages differentiated into endoderm and day 15 progenitors with similar efficiencies (Supplementary Fig. 2).

Over the course of 3 weeks, these cultures grew from single cells into self-organizing 3D structures. In order to define the transcriptional identity of these organoids, we performed RNA sequencing of bBU1c2 at day 42 of differentiation, comparing the CD47$^{hi}$ (enriched for NKX2-1-expressing cells) as well as the CD47$^{lo}$ (enriched for non NKX2-1-expressing cells) outgrowth to day 8 of differentiation. As shown in Fig. 1b, the CD47$^{hi}$ cells sorted on day 15, re-plated in 3D Matrigel in CK + DCI, and analyzed on day 42, were enriched for expression of transcripts encoding typical lung markers such as $NKX2$-1, as well as markers of type 2 alveolar cells including $SLC34A2$, $NAPSA$, $LPCAT1$, $SFTPC$, and $SFTPB$. In contrast, organoids generated from the day 15 CD47$^{lo}$ outgrowth expressed genes of mixed tissue identity, including small intestine ($CDX2$, $LYZ$, and $CDH17$), colon ($SATB2$ and $CEACAM5$), and liver ($SERPINA1$ and $HNF4\alpha$). When analyzed using Enrichr[28,29] (referenced to the human gene atlas), the number one hit in 'Cell Type' for the top 350 significantly upregulated genes in the CD47$^{hi}$ outgrowth was fetal lung, while the top hits for the CD47$^{lo}$ outgrowth were colon and small intestine (Fig. 1c). The data also showed that Vimentin ($VIM$), a mesenchymal marker, was significantly downregulated, while the canonical epithelial marker EpCAM, was significantly upregulated, in the CD47$^{lo}$ sorted cells (see below).

Whole mounts of BU3NGST GFP-derived organoids at day 85 stained for Cdx2 and the intestinal brush border component Villin showed robust 3D epithelial organoid formation, containing a significant number of Cdx2/Villin co-expressing cells (Fig. 1d). In addition, the $NKX2$-1$^+$ cells grew into spheres comprised of type II alveolar epithelial cells, as previously described[13].

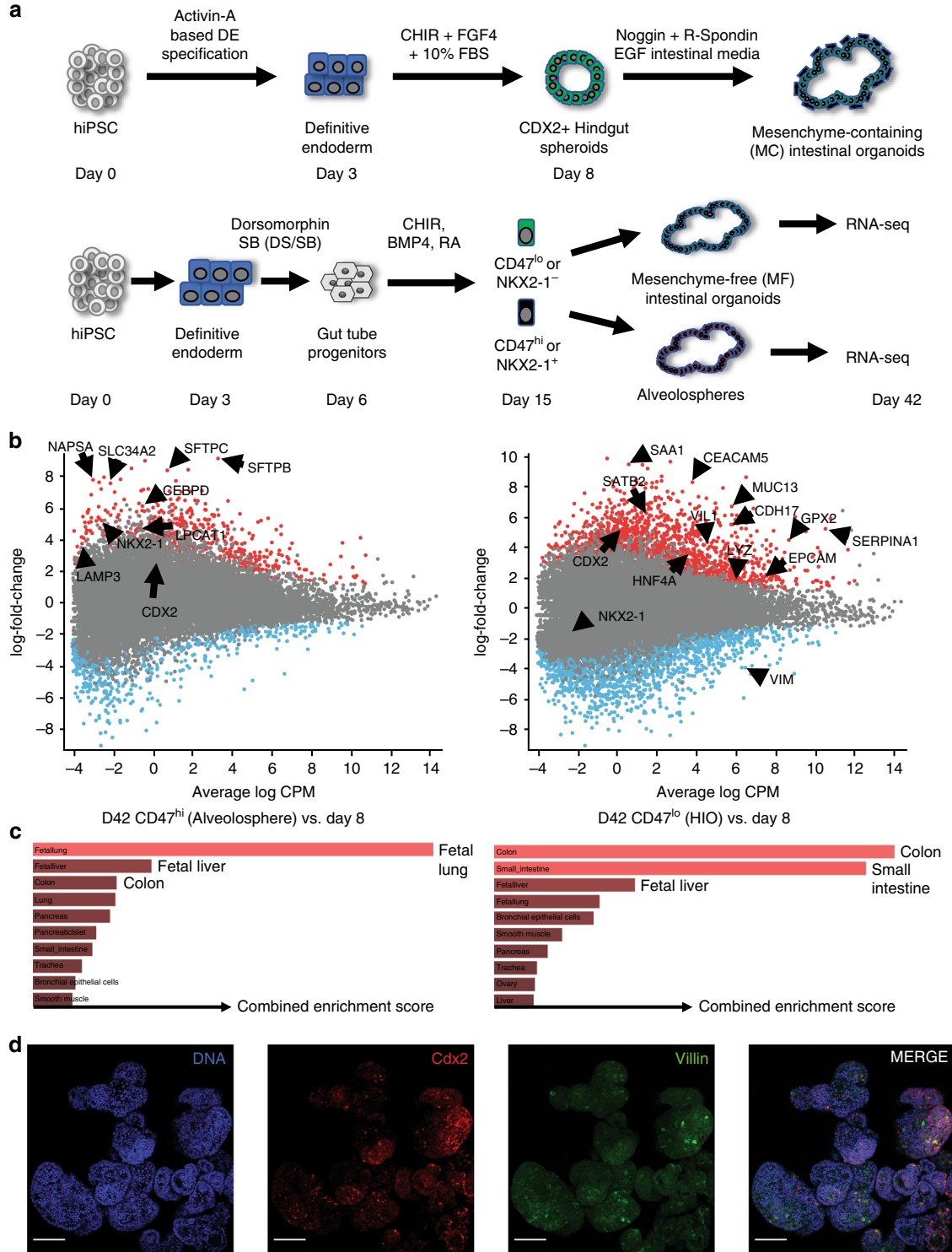

**Fig. 1 Emergence of intestinal-competent progenitors from iPSCs. a** Schematic of comparison between mesenchyme-containing (MC) HIO vs mesenchyme-free (MF) directed differentiation protocols. **b** Mean Average (MA) Plots of significantly differentially expressed genes that were either upregulated (red dots) or downregulated (blue dots) in digital gene expression analysis from day 42 (D42) organoids sorted for CD47 on day 15, comparing the CD47hi (Alveolospheres, left) and CD47lo (HIOs, right) cultured in CK-DCI, as compared with day 8 (D8) progenitors (p-value = <0.05 calculated as described[82]). **c** Gene set enrichment analysis using Enrichr analyzing the top tissue types when referenced to the human gene atlas. Length of red bars indicates combined enrichment score. All bars are adjusted p-value = <0.05 calculated as described[28,29]. **d** Representative micrographs of whole mounts of day 85 organoids derived from BU3NGST NKX2-1GFPneg sorted outgrowth demonstrate colocalized expression of Cdx2 and Villin (VIL) (scale bar = 50 μm, representative of n = 3 differentiations).

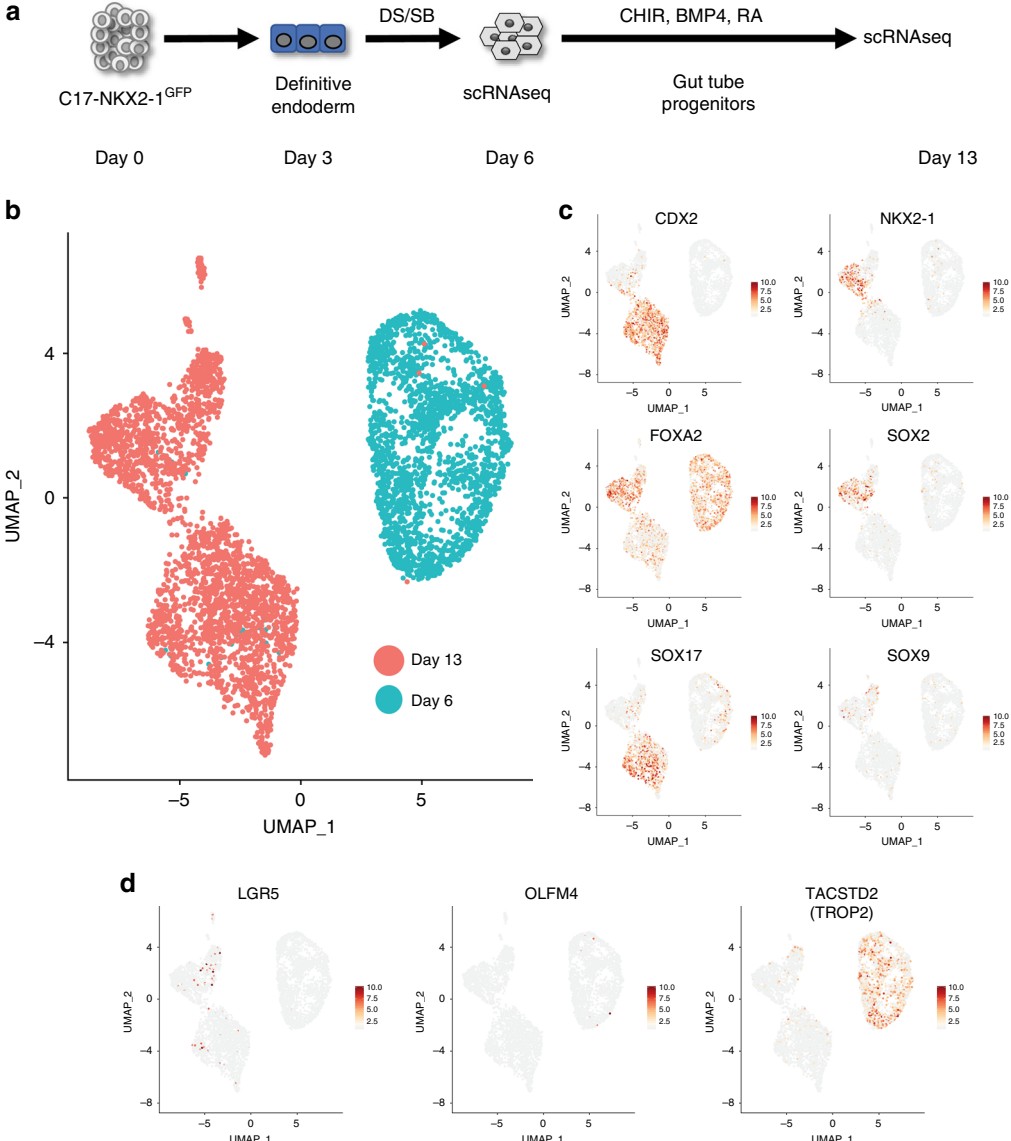

**Fig. 2 scRNAseq of day 6 and day 13 progenitors. a** Experimental schematic of scRNAseq, performed at day 6 (following 3 days of dual-smad inhibition) and day 13 (following 7 days of specification in CBRa). **b** UMAP visualization of cells at days 6 and 13 of differentiation. **c** UMAP visualization of expression of specific endodermal (*SOX17* and *FOXA2*), lung (*NKX2-1*, *SOX2*, and *SOX9*) and intestinal (CDX2) markers at days 6 and 13 of differentiation. Color scale indicates normalized log fold change of gene expression. **d** UMAP visualization of expression of intestinal stem cell markers *LGR5*, *OLFM4*, and *TACTSTD2* (TROP2) in cells at days 6 and 13 of differentiation.

**scRNAseq captures the emergence of intestinal progenitors.** The use of dual-SMAD inhibition was originally reported to strongly induce anterior foregut specification in pluripotent stem cell-derived endoderm[23]. To further understand progenitor cell lineage commitment at single-cell resolution early on in differentiation, we performed single-cell mRNA sequencing (scRNAseq) as depicted in Fig. 2a. We differentiated the C17 *NKX2-1-GFP* (C17)[30] iPSC line as described above, and performed scRNAseq using the 10x Chromium platform at day 6 (after 3 days of dual-smad inhibition) and day 13 (after 7 days of CHIR and BMP4 stimulation) of directed differentiation. At day 6, we analyzed 2215 cells, at a depth of 53,297 reads per cell, while at day 13 of differentiation, 2763 cells were analyzed with 53,471 reads per cell. We then performed dimensionality reduction as visualized using uniform manifold approximation and projection (UMAP), depicting the day 6 and 13 cells in the same plot (Fig. 2b). Unsurprisingly, these two populations clustered independently from one another in an unsupervised manner,

indicating the major transcriptional changes in cell identity that are known to occur during the early stages of directed differentiation.

We sought to look at a subset of genes that mark endoderm (*SOX17* and *FOXA2*), anterior foregut (*SOX2*), as well as intestinal (*CDX2*) and Lung/Thyroid (*NKX2-1*) progenitors[31,32]. The dorsal and ventral foregut endoderm are marked by *SOX2* and NKX2-1, respectively[33,34], while the boundary of *SOX2* and *CDX2* expression in developing endoderm marks the foregut from the posterior endodermal tissues[32]. After 3 days of endodermal specification, followed by 3 days of dual-smad inhibition, the anterior marker *SOX2* was expressed in only a subset of cells at day 6 (Fig. 2c). In addition, by day 13 a significant number of cells expressed *CDX2* and *SOX17*, which notably do not overlap with the *NKX2-1*-expressing cells. Furthermore, the pancreatic master regulator *SOX9* is not widely expressed in day 13 cells (Fig. 2c), suggesting that our progenitor population at day 13 does not contain a significant proportion of

pancreas-competent cells[35]. We also sought to examine the expression of intestinal stem cell markers early on in directed differentiation. Figure 2d demonstrates that there are a significant number of cells at day 6 that express *TROP2* and some expressing *OLFM4*, while a subset of cells at day 13 express *LGR5*. Overall, these data support the presence of a large *CDX2*[+] progenitor population with the potential to give rise to HIOs.

**Proximal specification of intestinal progenitors**. Having demonstrated that the NKX2-1[−] population contains gut-competent progenitors, we then sought to identify culture conditions that favor the emergence of regionally-patterned populations of intestinal-specific organoids. Day 15 BU3NGST cells were first sorted to isolate GFP negative cells, and cultured in a range of media conditions, including the previously published Noggin/R-Spondin-based intestinal media (IM)[15], as well as CK + DCI, and a variety of combinations of FGF4, KGF and Wnt activation (Fig. 3a). The intestinal media with CHIR and KGF (IM + CK) was shown to generate more organoids than any other media condition (Fig. 3b, c). In an attempt to ascertain regional identity and to further characterize the transcriptional profile of these organoids, quantitative real-time PCR (qRT-PCR) was performed for a variety of genes (Fig. 3d). *CDX2*, intestinal-specific cadherin *CDH17*[36], and *VIL1* (Villin)[37] were all highly expressed as compared with undifferentiated iPSCs in CK + DCI and IM + CK, as well as in the previously published intestinal medium[15] supplemented with CHIR (enabling us to discern the effect of CHIR from KGF). Notably, these markers were expressed at similar levels compared with a primary control (Fig. 3d). However, *PDX1*, a homeobox transcription factor essential for duodenal and pancreatic development[38], as well as *GATA4*, another proximal small intestinal marker, were significantly upregulated in the IM + CK condition as compared with both the CK + DCI organoids as well as adult colon (Fig. 3d). Notably, the IM + CK organoids had significantly lower levels of *SATB2*[20], Albumin (*ALB*), and Pepsinogen C (*PGC*)[39] expression compared with the CK + DCI HIOs (and IM + CHIR), suggesting that the IM + CK condition results in HIOs that are more homogenous and express markers specific to intestinal lineages, while preventing the emergence of hepatic and gastric lineages (Supplementary Fig. 3). The CK + DCI HIOs expressed significantly more *SATB2*, a colonic marker, than either the IM + CK or IM + CHIR HIOs (Fig. 3d). Organoids grown in all conditions express high levels of lysozyme, an antimicrobial protein expressed by Paneth cells throughout the GI tract[40]. Expression of the intestinal markers Cdx2 and Villin was also confirmed by immunohistochemistry (Fig. 3e), further validating that the IM + CK conditions generated the most robust intestinal-specific organoids.

**Generation of a CDX2-GFP reporter iPSC line BU1CG**. We demonstrated that both NKX2-1[−] and CD47[lo] sorted cells are enriched for intestinal progenitors that have the potential to grow into *CDX2*[+] organoids that express a variety of markers specific for intestinal epithelium. However, in order to identify, profile, and purify putative *CDX2*[+] intestinal progenitors throughout each stage of directed differentiation, we targeted the *CDX2* locus with an *eGFP* fluorescent reporter, generating a *CDX2-GFP* knock-in reporter cell line (Fig. 4a and Supplementary Fig. 4). Using CRISPR/Cas9, we gene-edited a normal iPSC line, bBU1c2[27], using a synthesized self-linearizing DNA oligonucleotide containing a 2A-eGFP-polyA flanked by two 400 base pair homology arms as a donor and a Cas9-GFP plasmid, eliminating the need for subsequent selection marker excision (see Methods). Due to the presence of a self-cleaving 2A peptide and a

targeted insertion site just upstream of the endogenous *CDX2* stop codon, the *CDX2* gene was not inactivated as a result of gene editing (Fig. 4a). PCR confirmed that the construct was inserted into the desired locus in 70% of the clones picked from one of the two sgRNA's used (Fig. 4b Clone 109, hereafter referred to as BU1CG). Off target screening for the top three most likely off target gene insertion sites (*NHLRC4*, *RAI4*, and *SPP3*) based on the sgRNA sequence revealed no aberrant indels (Supplementary Fig. 4c).

The selected BU1CG line showed a stable iPSC morphology upon passage and normal karyotype (Supplementary Fig. 4d), and was subsequently differentiated into definitive endoderm, yielding on average 76.7% CXCR4/c-KIT double positive cells ($n = 7$, representative flow cytometry in Supplementary Fig. 5a). Cells were then differentiated into intestinal progenitors using the MF protocol, and sorted for *CDX2*[GFP] at day 15 of differentiation, and then plated as single cells in 3D Matrigel droplets, and incubated in multiple culture conditions similar to the approach outlined for BU3NGST (Fig. 4c). At day 15, we generated an average of $1.377 \times 10^7$ cells per input well of iPSCs containing $2 \times 10^6$ cells ($n = 3$ differentiations, Supplementary Fig. 5b). We confirmed the fidelity of the reporter by sorting cells at day 15 based on GFP and as expected, expression of *CDX2* tracked with the GFP[+] sorted cells (Fig. 4d). Taking advantage of the reporter, we followed the emergence of these putative intestinal progenitors based on *CDX2*[GFP] expression. As shown in Fig. 4e, *CDX2*[GFP] positive cells began to emerge at day 8 of differentiation, and by day 13 they represented $41.166 \pm 20.53\%$ ($n = 6$, mean ± s.d.) of all cells in the culture. Staining of mature HIOs at day 40 further confirmed the fidelity of the GFP reporter, depicting nuclear Cdx2 staining colocalizing with cytoplasmic GFP (Fig. 4f).

Confirming our previous findings, IM + CK and CK + DCI generated significantly more *CDX2*[GFP] organoids per input cell as compared with the other combinations of intestinal medium, Wnt, and FGF4/KGF stimulation (Fig. 4g, h). Immunofluorescence and light microscopy analyses of the resulting organoids revealed luminal, organized multicellular structures with high *CDX2*[GFP] expression (Fig. 4i, j). In contrast, sorting GFP negative cells for re-plating and further outgrowth in the same IM + CK conditions resulted in almost complete depletion of gut-competent cells with poor outgrowths containing significantly fewer GFP+ cells ($0.49\% \pm 0.052$, $n = 3$ mean ± s.d.). Not surprisingly, when the GFP negative cells were cultured in CK + DCI conditions the outgrowth showed high expression of the lung marker, *NKX2-1* (Supplementary Fig. 5c), most of them negative for *CDX2*[GFP] (Supplementary Fig. 5d). These data provide strong evidence for the early emergence of putative intestinal progenitors during the MF protocol and suggests that most, if not all intestinal capacity resided in the *CDX2*[GFPpos] population, as the *CDX2*[GFPneg] cells failed to form robust 3D structures when cultured in the IM-CK condition (Fig. 4i, left inset).

**iPSC-derived HIOs grow in the absence of mesenchymal support**. Intestinal organoids grown from intestinal crypts can self-sustain their in vitro expansion in the absence of mesenchymal support[2,41], something that has yet to be recapitulated with iPSC-derived organoids. As mentioned above, previously reported iPSC-derived intestinal directed differentiation protocols also lead to the generation of Cdx2 negative mesenchyme, which was proposed to secrete a variety of factors that induce and support the growth of the intestinal epithelium[42]. We sought to determine if the HIOs obtained using our protocol, in fact, differentiated in the absence of mesenchymal support (Fig. 5a) compared side by side with the previously described protocol[15]. Light/Fluorescence

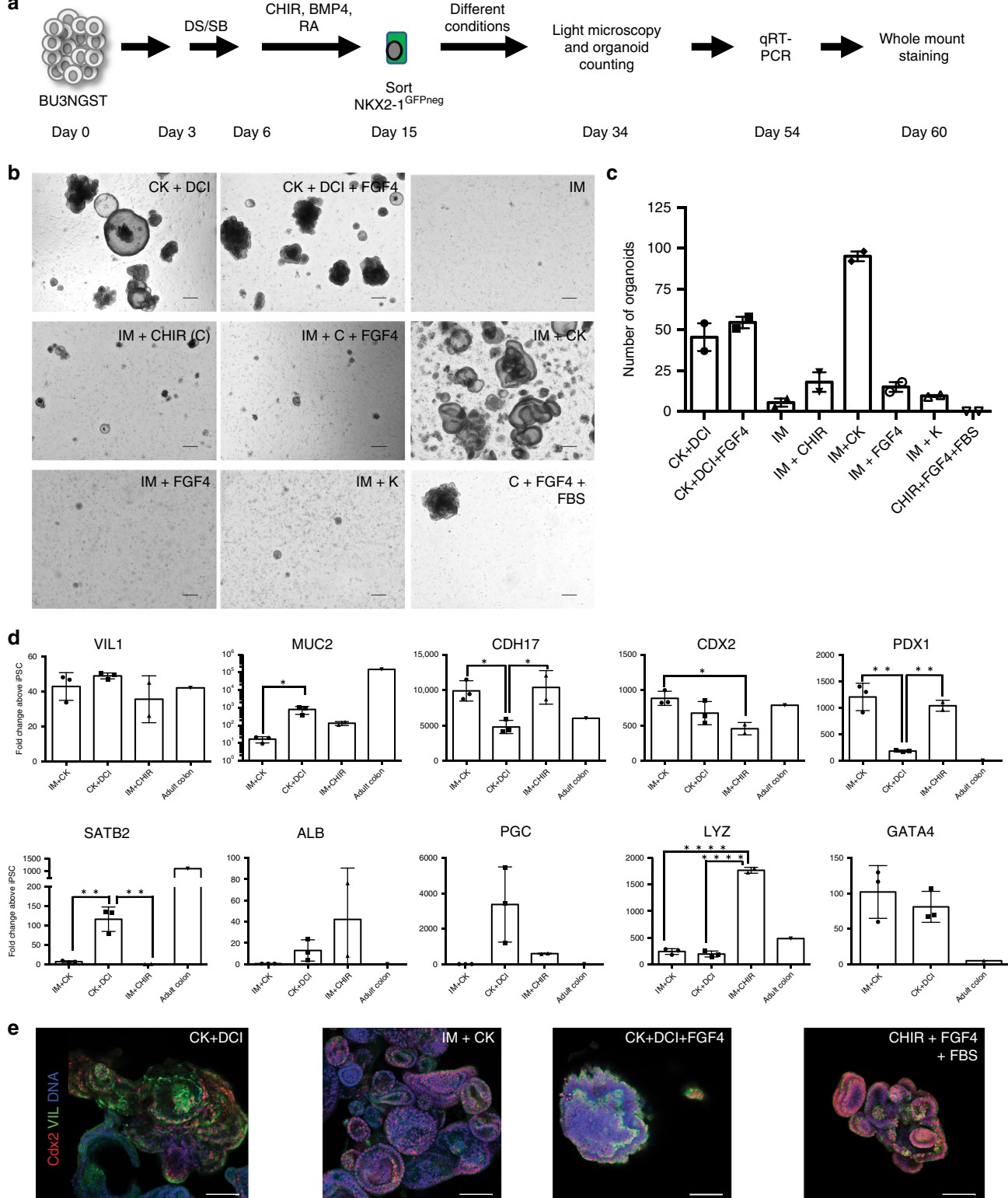

**Fig. 3 Proximal small intestinal specification following dual-smad inhibition. a** Experimental Schematic of directed differentiation. **b** Representative micrographs of cells grown in different media conditions at day 34 of differentiation after sorting for NKX2-1$^{GFP}$ negative cells at day 15 (scale bar = 300 μm, representative of n = 2 independent wells per condition). **c** Quantification of number of organoids per well at day 50 in different culture conditions. Error bars represent s.e.m. from n = 2 independent wells per condition. **d** qRT-PCR from day 54 HIOs cultured in either IM + CK, CK + DCI, or IM + CHIR normalized to day 0 hiPSCs and compared with a primary control (human adult colon) (2$^{-\Delta\Delta CT}$, technical triplicates normalized to *GAPDH* or *ACTB* (β-ACTIN), n = 3 independent differentiations except IM + CHIR (IM + C, n = 2 independent differentiations). Error bars represent the s.d., statistical significance, where indicated, determined by one way-ANOVA followed by Tukey test, *p < 0.05, **p < 0.005, ****p < 0.0001). **e** Whole mount immunofluorescence of day 60 organoids stained for Villin (VIL), Cdx2 and DNA (blue) (scale bar = 100 μm, representative of n = 3 differentiations).

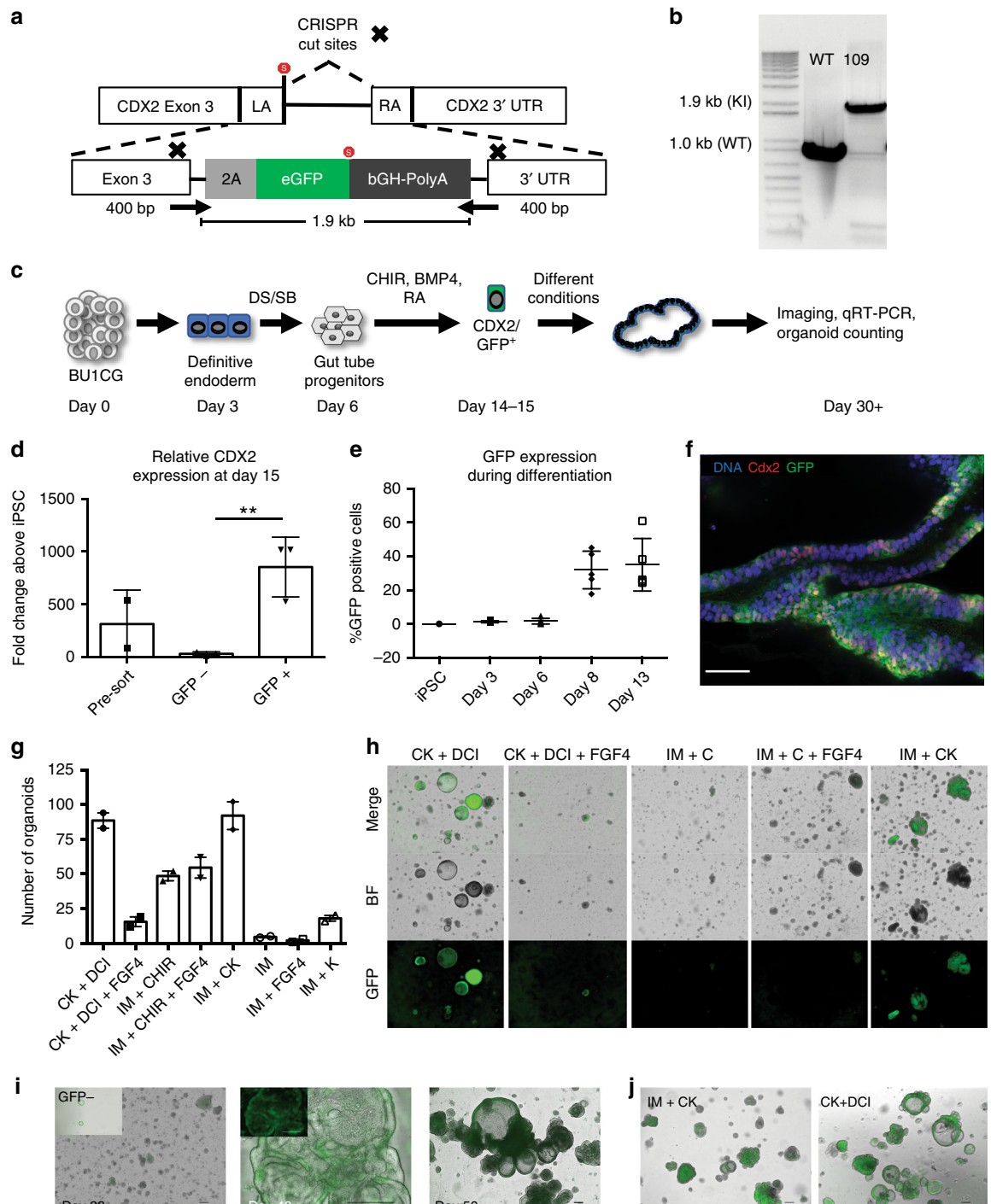

**Fig. 4 A *CDX2-GFP* iPSC reporter line for intestinal differentiation. a** Detailed schematic of the reporter construct and PCR screening primer sites (arrows). **b** Positive PCR screening of mono-allelic (17) and bi-allelic (109) knock-in clones derived from iPSC line bBU1c2. **c** Experimental schematic of differentiation of BU1CG into HIOs. **d** qRT-PCR for *CDX2* expression in cells at day 15 of differentiation comparing sorted GFP⁺ to both GFP⁻ sorted cells as well as pre-sort ($2^{-\Delta\Delta CT}$ technical triplicates normalized to *GAPDH*, $n = 3$ independent sorts, error bars represent the s.d., $p = 0.007$, **$p < 0.01$) as determined by unpaired Student's *t*-test). **e** FACS analysis of GFP expression during the first two weeks of differentiation ($n = 4$ independent differentiations, error bars represent the s.d.). **f** Quantification of number of organoids per independent well obtained at day 50 of differentiation (error bars indicate s.e.m. from $n = 2$ independent wells per condition). **g** Whole mount immunofluorescence of HIOs at D40 of differentiation stained for Cdx2, GFP, and DNA (blue) (scale bar = 50 μm, representative of $n = 6$ organoids from $n = 2$ differentiations). **h** Representative micrographs of HIOs at day 34 of differentiation using different media conditions (originally sorted at day 14 for *CDX2^GFP*, scale bar = 100 μm, representative fields of view of $n = 2$ wells per condition). **i** Representative micrographs showing the formation of HIOs from BU1CG after sorting GFP + cells at day 14 (scale bar = 200 μm). Inset shows limited outgrowth from GFP− sorted cells cultured in the same media conditions (IM + CK) (scale bar = 200 μm) (representative of $n = 3$ differentiations). **j** Representative micrographs showing HIOs at day 45 comparing CK + DCI vs IM + CK conditions (scale bar = 200 μm, representative of $n = 6$ differentiations).

microscopy demonstrated the presence of $CDX2^{GFPpos}$ HIOs differentiated using both the MC and MF protocols (Fig. 5b). However, there were clear morphologic differences, highlighted by the presence of GFP negative cells surrounding the $CDX2^{GFP}$ positive epithelium in the MC organoids (Fig. 5b, top). Staining for Vimentin revealed that these GFP negative cells were mostly positive for Vimentin which were not present in the MF protocol (Fig. 5c), findings that were confirmed by RNA-Seq (Supplementary Fig. 3). Representative flow cytometry for the epithelial-specific marker EpCAM (Fig. 5d) demonstrated that in the MF protocol virtually all cells were epithelial, in contrast to the MC protocol where up to 50% of the cells were EpCAM negative.

Furthermore, we decided to investigate whether the ability of the HIOs to grow in the absence of mesenchyme was established during emergence of $CDX2^{GFP}$ positive cells regardless of which protocol we used. Serial flow cytometry for EpCAM over the early stages of differentiation demonstrated the maintenance of EpCAM expression in the MF protocol and gradual loss of EpCAM expression in the MC protocol (Fig. 5f). In addition, scRNAseq of cells at days 6 and 13 (as described in Fig. 2) demonstrated that there is widespread expression of EpCAM in cells at those time points of differentiation in the MF protocol, supporting our flow cytometry findings (Fig. 5g).

Using this same dataset, we also tracked expression of mesenchymal and mesodermal markers at these time points, including VIM, COL1A1, COL3A1, FN1, THY1, and ACTA2 (Supplementary Fig. 6a). With the exception of VIM and FN1, the vast majority of cells at day 13 did not express the other markers listed above. It has been reported that epithelial cells express mesenchymal genes including VIM and FN1 during early organogenesis (E9-11.5)[43,44], which may explain why our day 13 cells express these markers.

Finally, we compared the outgrowths of hindgut obtained via the first 8 days of the MC protocol[15] to CDX2 positive cells obtained at day 15 of the MF protocol cultured in 3D Matrigel® in several media conditions. As shown in Fig. 5e, HIOs derived from $CDX2^{GFPpos}$ sorted cells contained significantly more EpCAM + cells (99.25 ± .49%, $n = 6$ mean ± s.d.) compared with MC differentiated cells (49.92 ± 29.9%, $n = 6$ mean ± s.d.), regardless of the media. In order to interrogate whether the day 15 FACS step of the MF differentiation is responsible for mesenchymal depletion, we also performed an MF differentiation that omitted the day 15 sort, and plated progenitors into 3D culture conditions at day 15. At day 30, 88.6% of cells were EpCAM$^+$ by flow cytometry (Supplementary Fig. 6b), supporting the notion that this differentiation protocol enables the emergence of intestinal organoids without the need for mesenchymal support.

**MF HIOs contain a variety of intestinal epithelial cell types**. The intestinal epithelium is made up of a diverse group of cell types each occupying a specific functional niche. These include absorptive enterocytes, secretory Paneth cells that secrete Lysozyme (LYZ), enteroendocrine cells that express Chromogranin A (CHGA), and mucin-secreting goblet cells, among others[45]. In order to assess whether our HIOs contained cell types present in mature intestinal epithelium, we performed immunohistochemistry comparing the organoids grown in CK-DCI vs. IM-CK conditions (Fig. 6a–c). We observed the presence of enteroendocrine cells (positive for chromogranin A, Fig. 6a) and Paneth cells (positive for Lysozyme, Fig. 6b), in the context of organoids comprised mostly of Villin-expressing putative enterocytes, at the expected densities for these particular cell types in both conditions. We also noticed that the IM + CK organoids were largely negative for colonic mucin Muc2, while the CK +

DCI HIOs showed robust staining for luminal Muc2 (Fig. 6c), suggesting that the IM-CK conditions promote a more proximal identity in contrast to the CK-DCI conditions. In order to further investigate this, we performed qRT-PCR for a panel of genes (Fig. 6d) specifically expressed by proximal or distal intestinal epithelium. Indeed, the gene expression profile of the HIOs obtained through these two different conditions strongly supported our previous observations that the CK-DCI protocol gives rise to heterogeneous organoids containing many cells expressing colonic markers, as well as markers of several GI tract tissues, particularly when compared with RNA from a primary control from an adult human colonic tissue sample. In contrast, the IM-CK conditions promote the emergence of HIOs with a more defined identity corresponding to proximal small intestine/duodenum. These HIOs express high levels of CDX2, LYZ, CDH17, GATA4, and PDX1, while expressing significantly lower levels of the distal colonic markers SATB2 and MUC2.

**Patient HIOs are suitable for disease modeling**. While the translational potential of organoids to the ultimate goal of cell or tissue replacement therapy is still years away, they have already demonstrated significant utility in disease modeling, particularly in the context of monogenic disorders such as Familial Adenomatous Polyposis (FAP)[46] and cystic fibrosis (CF)[12,47,48]. In CF, cell-based models have proved invaluable in developing CFTR modulators. Primary airway epithelial cells are the gold-standard for predicting CFTR drug-responsiveness however significant progress has been made using rectal organoids to theratype CF patients[49–51]. Given that CFTR is highly expressed in the intestinal epithelium, we tested, in proof-of-concept experiments, the feasibility of measuring CFTR function in our iPSC-derived HIOs (see schematic, Fig. 7a).

In order to do so, it was first necessary to test whether iPSCs carrying CFTR mutations were capable of differentiating into HIOs using our MF protocol. For this purpose, we used our published cystic fibrosis (CF)-specific iPSC line, C17[11] and differentiated cells toward distal lineages using the CK + DCI condition. Similar to what was observed in previous differentiations with wild type cell lines, C17 was successfully differentiated into 3D HIOs expressing Cdx2 and Villin (Fig. 7b). qRT-PCR confirmed that differentiated HIOs expressed CFTR relative to undifferentiated cells, but at lower levels when compared with adult colon (Fig. 7c). To control for the potential effects of genetic background on CFTR measurement we used an iPSC line from an individual homozygous for the ΔF508 mutation (ΔF508). This ΔF508 iPSC line was previously gene-edited to correct the ΔF508 mutation in one allele (ΔF508-corrected)[30]. ΔF508, ΔF508-corrected and WT HIOs were differentiated until day 30, as above. We applied methodologies including steady-state lumen area (SLA) and forskolin-induced swelling (FIS), previously developed for the analysis of CFTR function in rectal organoids[48,51]. At baseline the average organoid size was significantly smaller in ΔF508 compared with WT and ΔF508-corrected HIOs (Fig. 7d). SLA was also significantly lower in ΔF508 compared with ΔF508-corrected HIOs (Fig. 7e and Supplementary Fig. 7b). In response to FIS, WT HIOs started to swell within 30 min (Supplementary Fig. 7a). After 24 h of forskolin, no significant change in whole well cross-sectional area (CSA) was detected in ΔF508 HIOs (mean CSA 1.073 ± 0.04702); whereas ΔF508-corrected and WT HIOs significantly increased in CSA (2.190 ± 0.3051 in the ΔF508-corrected and 2.190 ± 0.1950 in the WT, $n = 3$ independent differentiations per cell line, data represents mean ± s.d.) (Fig. 7f, g, Supplementary Movies 1–6, representative of $n = 3$ differentiations and $n = 3$ wells per condition).

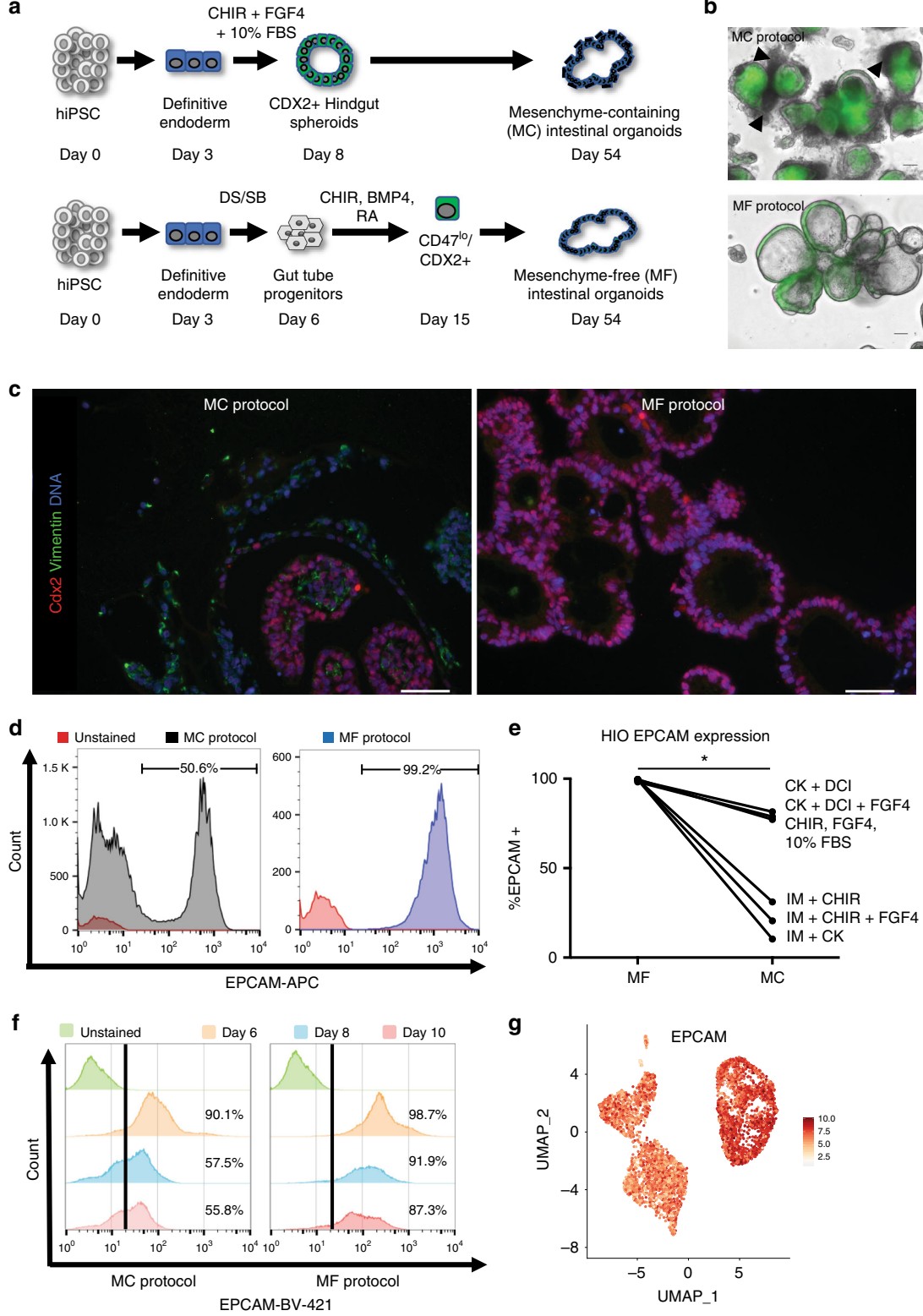

**Fig. 5 iPSC-derived HIOs grow in the absence of mesenchymal support. a** Experimental schematic of MF and MC directed differentiations. **b** Light microscopy representative micrographs of merge images from BU1CG-derived HIOs cultured under MC vs MF conditions (scale bar = 100 μm, representative of $n = 3$ differentiations). **c** Representative fluorescent micrographs of HIOs derived using the MC vs MF Protocol stained for the mesenchymal marker Vimentin along with Cdx2 (scale bar = 50 μm, representative of $n = 5$ organoids from $n = 3$ differentiations). **d** Flow Cytometry of single-cell suspensions from HIOs differentiated using the MC vs MF protocol stained with the epithelial marker EpCAM. **e** Comparison of the % of EPCAM+ cells as measured by flow cytometry in HIOs cultured in different media conditions at day 54 of differentiation (paired Student's $t$-test, *$p <$ 0.05). **f** Flow cytometry for EpCAM expression of cells at days 6, 8, and 10 of both MC and MF differentiations. **g** UMAP representation of EpCAM expression at days 6 and 13 of differentiation by sc-RNAseq.

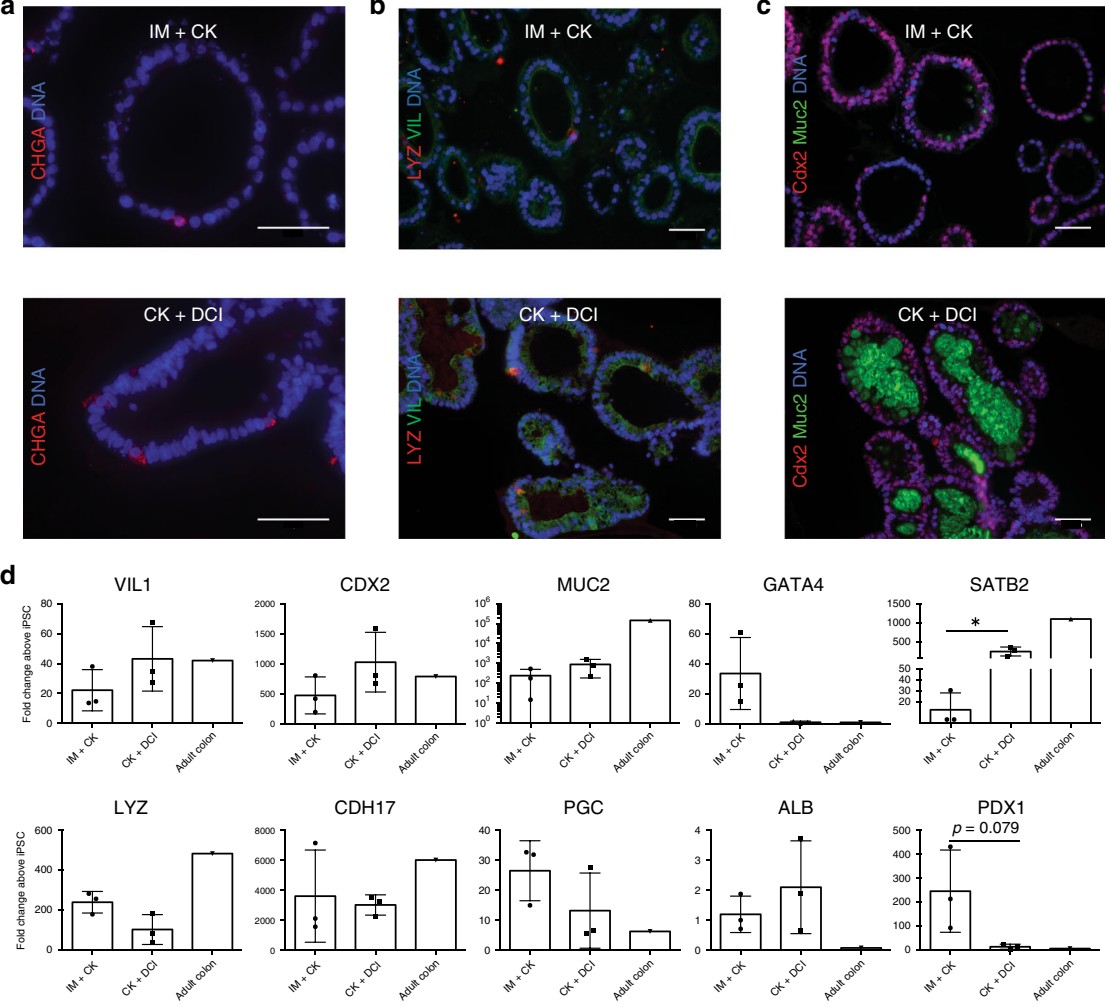

**Fig. 6 CDX2$^{GFP+}$ sorted organoids show conserved regional specificity. a–c** Sections of paraffin embedded organoids cultured in either IM + CK or CK + DCI stained for Chromogranin A (CHGA), Lysozyme (LYZ), Villin (VIL), Cdx2, and colonic mucin Muc2 (scale bar = 50 μm, representative of $n = 5$ organoids from $n = 2$ differentiations). **d** qRT-PCR for various genes of interest, from D54 HIOs cultured in either CK + DCI or IM + CK as compared with a primary control normalized to day 0 hiPSCs ($2^{-\Delta\Delta CT}$, technical triplicates normalized to *GAPDH* or *ACTB* (β-ACTIN), $n = 3$ differentiations, error bars represent the s.d., statistical significance where indicated determined by unpaired Student's *t*-test, *$p < 0.05$).

## Discussion

The original description of the use of dual-smad inhibition by Green et al.[23] to induce differentiation of anterior foregut from endoderm-patterned pluripotent stem cells suggested that these conditions particularly suppressed *CDX2* expression in endodermal cells. Our findings show that, although lung competence is induced in a subset of cells, many are, in fact, not patterned toward anterior foregut endoderm following 72 h of dual-smad inhibition. Indeed, our scRNA-seq data showed that there is significant heterogeneity throughout directed differentiation (Fig. 2b, c), and by day 13, a significant number of cells were *CDX2*$^+$ (Fig. 2c). This result was confirmed using our BU1CG line that showed robust upregulation of *CDX2*$^{GFP}$ starting at day 8 of differentiation (Fig. 4). A potential explanation for this discrepancy might lie in the fact that most of the data in the Green et al. studies were obtained at earlier time points during differentiation using a single human embryonic stem cell (hESC) line. Their original protocol relied on the use of embryoid bodies as a starting point, which could generate mesodermal derivatives that may affect the outcome of the differentiation. Our data suggests that although dual-smad inhibition can facilitate the emergence of progenitors with anterior foregut capacity, it does not prevent the differentiation of a robust progenitor

population capable of specifying into many endodermal lineages including more posterior gut tube derivatives, such as small and large intestine, liver, pancreas, and stomach, primarily as a result of strong putative activation of the Wnt/β-Catenin pathway due to treatment with GSK3β inhibitor CHIR99021. It is important to acknowledge that at this point we cannot rule out additional Wnt-independent effects of CHIR99021. While the original protocol for the generation of HIOs from human iPSCs employed Wnt3A as the primary Wnt agonist to promote intestinal progenitor specification[15], many subsequent manuscripts have described directed differentiation of iPSCs to HIOs that are reliant on CHIR99021 for Wnt/β-Catenin activation[8,20,52–54]. In addition, CHIR99021 has been well characterized as a strong inducer of the Wnt/β-Catenin signaling pathway[55].

It was intriguing to find that cells sorted based on *CDX2*$^{GFP}$ (or CD47/*NKX2-1*) at day 15 of differentiation and cultured in the same media conditions gave rise to very different cell lineages. Indeed, sorted CD47$^{hi}$ (or *NKX2-1*$^+$) cells vs CD47$^{lo}$ (or *NKX2-1*$^-$) cells cultured in CK + DCI gave rise to very different organoids, i.e., alveolar-like vs posterior lineages (gut, liver, stomach), respectively. Whether or not this results from a true cell fate decision and commitment to an anterior vs posterior fate established early during

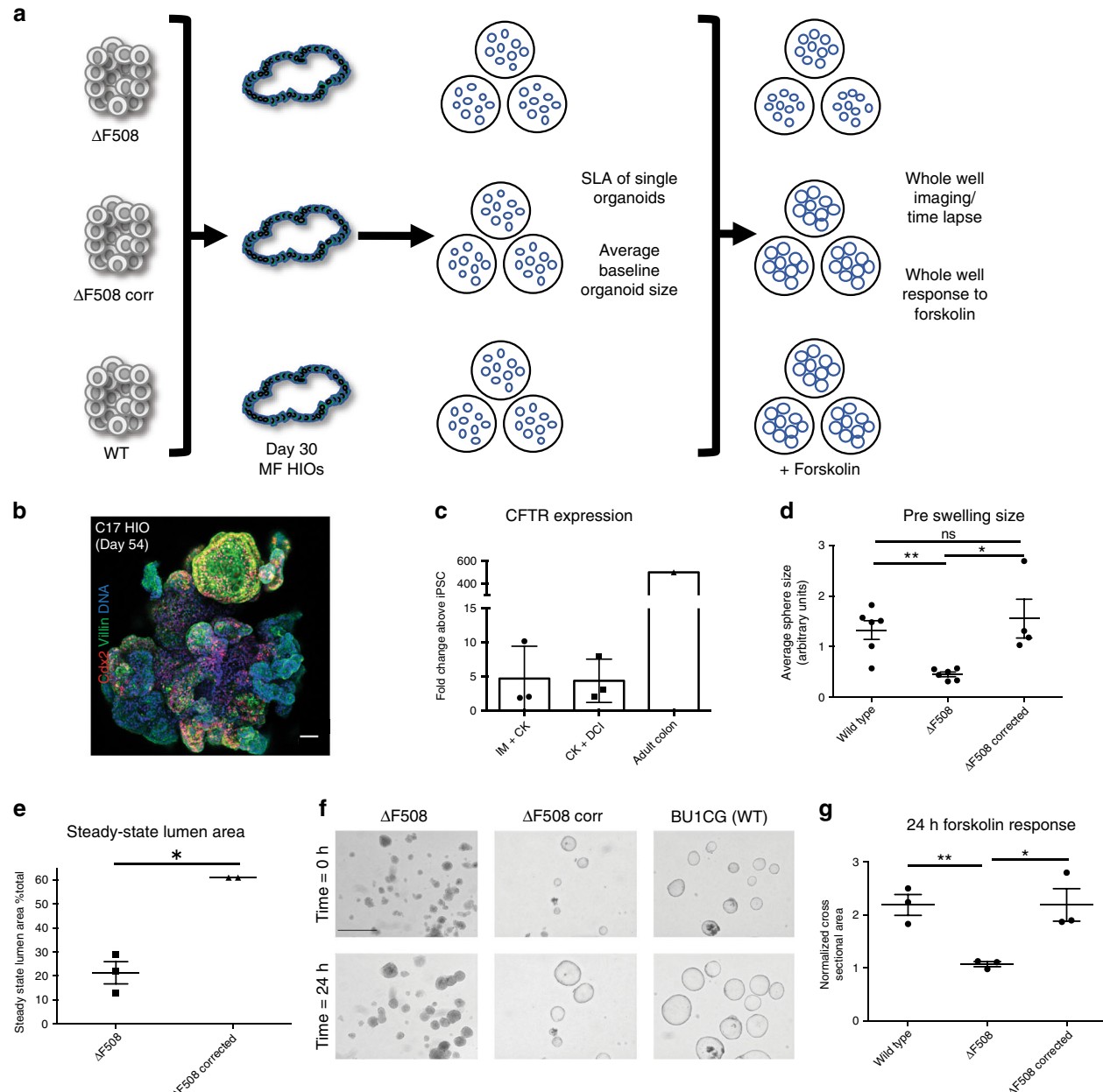

**Fig. 7 Patient specific mesenchyme-free-derived HIOs are suitable for disease modeling. a** Schematic overview of experiment to measure CFTR function in ΔF508, ΔF508-corrected and WT HIOs ($n = 3$ independent differentiations per iPSC line). **b** Representative micrograph of whole mount of distally patterned HIOs generated from the C17 iPSC line at day 54 of differentiation, sorted for $NKX2\text{-}1^{GFP-}$ at day 15, and cultured in CK + DCI (scale bar = 50 µm, representative of $n = 2$ differentiations). **c** qRT-PCR for *CFTR* expression in HIOs cultured in both IM + CK and CK + DCI at day 54 of differentiation ($2^{-\Delta\Delta CT}$, technical triplicates normalized to *GAPDH*; $n = 3$). **d** Average baseline organoid size of WT, ΔF508 and ΔF508-corrected HIOs ($n = 192$ mean spheres analyzed per iPSC line). **e** Quantification of the steady-state lumen area (SLA) in ΔF508 and ΔF508-corrected HIOs (see also Supplementary Fig. 7b). **f** Representative micrographs of HIOs pre- and post-24-h forskolin treatment (representative scale bar = 200 µm in upper left image, images represent $n = 3$ biological replicates per cell line and an average of $n = 8$ wells per replicate). **g** Quantification of change in whole well CSA in response to 24-h forskolin treatment (normalized to $T = 0$ h whole well CSA). Statistical significance where indicated determined by unpaired Student's *t*-test, *$p < 0.05$, **$p < 0.005$ ($n = 3$ independent differentiations per cell line, error bars represent the s.d.).

differentiation (accomplished via establishment of lineage specific epigenetic marks as shown in other systems[56–58]) is currently unknown and merits further investigation.

The fact that addition of KGF to the intestinal media generated the most robust conditions for proximal intestinal specification (IM + CK), with higher numbers of intestinal organoids is unsurprising, particularly given the role KGF has been shown to play in vivo in the intestinal epithelium. At the time of its original discovery in 1989, human KGF, which was ultimately re-classified as fibroblast growth factor 7 (FGF7), was shown to exert powerful paracrine effector functions on epithelial cell growth[59]. Since then, others have reported that KGF plays a direct role in epithelial proliferation across a range of cell types in the GI tract, including multiple epithelial cell lines[60,61], as well as goblet cells[62]. Furthermore, increased KGF activity has been associated with epithelial regeneration in inflammatory bowel disease, both in animal models[63] and human biopsy samples[64]. From a mechanistic point of view, KGF is known to be expressed

by the mesenchyme acting via FGFR2b present in the intestinal epithelia[65,66]. This is the opposite to FGF4 which is expressed by the epithelium and acts on receptors present in the mesenchyme[67]. This might explain the key difference between the MC protocol and ours, and the fact that we can achieve robust differentiation in the absence of mesenchymal support, whereas prior protocols stimulate concomitant mesenchymal outgrowth. The functional interaction of intestinal epithelium with intestinal mesenchymal lineages is certainly of interest. However, for certain questions, notably the investigation of epithelial intrinsic defects, the ability to study an epithelial-only culture system is a major advantage, and would diminish potential experimental noise generated from a mesenchyme-containing system. An epithelial-only model facilitates the simple measurement of epithelial intrinsic function or dysfunction in a reductionist framework.

Finally, we selected to study cystic fibrosis in our HIO model based on the suitability of these organoids to study the epithelial expression of CFTR in the intestine, in addition to the established assays to measure CFTR function using primary rectal organoids[48,68]. We determined, in proof-of-concept experiments, that MF HIOs provide an organoid-based read-out of CFTR function. It is worth noting that for the purposes of these experiments, organoids remained in the differentiation media containing cAMP. Thus, organoids with functional CFTR protein are activated at baseline and this likely dampens the magnitude of acute CFTR activation with forskolin. There are several preclinical, cell-based models to assess CFTR dysfunction and rescue[49]. Important future questions of the CFTR assay in HIOs include: (1) Do ΔF508 HIOs detect CFTR rescue in response to CFTR modulators?, (2) Does in vitro CFTR rescue in HIOs predict clinical efficacy?, and (3) How does the HIO platform compare to established platforms in terms of sensitivity and positive predictive value? Compared with routine rectal biopsies, it takes several months to first reprogram and subsequently differentiate iPSCs into HIOs. However, there are key areas where the unique properties of HIOs could be helpful. First, given that iPSCs are able to be expanded indefinitely, there is the potential to scale-up HIO production for screening purposes. Second, the ability to gene-edit iPSCs and differentiate these cells into multiple tissue types could overcome the genetic variability and overwhelming effect of the infected, inflammatory milieu that limits experiments using primary human tissue. An iPSC-based approach offers the potential to study key questions, including the role of genetic modifiers in the heterogeneity of CF phenotypes. At a molecular level, the iPSC system could be applied to determine cell-type and tissue-specific differences in the regulation of CFTR expression[69]. Nevertheless, despite significant progress in the development of increasingly potent CFTR modulators for residual function mutations[70,71], there remains a subset of individuals whose mutations result in little to no CFTR protein and who represent a major unmet therapeutic challenge. The optimal preclinical platform for these individuals remains to be determined and human, scalable, patient-specific cell-based platforms that express CFTR at higher levels than primary bronchial epithelial cells may prove helpful in drug development.

## Methods

**hiPSC generation and culture and expansion.** All parental hiPSC lines previously published by our group (bBU1)[27] and others (C17, dF508)[12,30] were derived from normal donors (bBU1) or an individual with a published compound heterozygote CFTR mutation (C17), and have been shown to have a normal karyotype (46XY). All lines were maintained in feeder-free conditions using mTESR®1 (StemCell Technologies), and passaged onto hESC Matrigel® (Corning cat. no. 354277) coated 10 cm, 6-well, 12-well, and 24-well tissue culture dishes (Corning) as per the manufacturer's instructions. All human subjects studies were performed under signed consent and approved by the Boston University Institutional Review Board (IRB), protocol H-32506.

**Cloning of *CDX2-eGFP* into a blunt ended cloning vector.** Using an approach outlined by Zhang and colleagues[72], we used a synthetic self-linearizing oligonucleotide construct (sequence provided upon request) as a donor without the need for subsequent selection marker excision. Oligonucleotide constructs for the donor, guide RNAs, and sequencing/screening primers were ordered from Integrated DNA Technologies (IDT). All sequencing reagents are listed in Supplementary Table 3. CRISPR Guide RNA sequence and target sites were selected using the CHOPCHOP[73,74] and MIT CRISPR Design Tools. The synthetic donor construct was cloned into the blunt ended cloning vector, pJET.2, using the CLONEJet PCR Cloning Kit (ThermoFisher cat. no. K1231). The SpCas9-2A-GFP (PX458) plasmid with cloning backbone for sgRNA was obtained from the Zhang Lab through Addgene (Addgene #48138)[75].

**Gene editing of bBU1c2.** Parental bBU1c2 iPSCs from one confluent six-well plate and 2 10 cm dishes were dissociated from their tissue culture vessels using ReLeSR (StemCell Technologies), and dissociated into single-cell suspensions. In total, $6 \times 10^6$ cells were nucleofected with 5 μg of guide plasmid DNA and 5 μg of donor plasmid DNA, and re-plated on fresh 10 cm Matrigel-coated dishes. The cells were nucleofected on an Amaxa™ 4D-Nucleofector™ using the Lonza Nucleofector P3 Primary Cell 4D-Nucleofector™ X Kit (V4XP-3024), as per the manufacturer's instructions, on the "HESCell H9" program.

**Screening and banking of BU1CG.** Two days after nucleofection, the cells were dissociated once more into single cells, and sorted for Cas9-GFP in order to select for cells that had been successfully nucleofected with the Cas9-sgRNA plasmid. Of the $4 \times 10^6$ cells sorted, 0.5% of them were GFP+. The cells were then plated at varying dilutions on 10 cm dishes, and allowed to grow into colonies, which were then mechanically picked using a p20 pipette and re-plated into one well of a 24-well plate containing warm mTeSR1® with 5 μM Y27632. Genomic DNA was extracted from 96 clones using the QIAamp DNA Mini Kit (QIAGEN cat. no. 51304), and screened by PCR using the Herculase II Fusion DNA Polymerase (Aligent cat. no. 600675, as per manufacturer's instructions) for successful donor construct insertion. Amplified DNA was visualized by gel electrophoresis using GelRed® Nucleic Acid Gel Stain (Biotium cat. no. 41002) and imaged using a BioRad GelDoc™XR System, along side the 1 Kb Plus DNA Ladder (ThermoFisher cat. no. 10787018). The gel presented in the main Fig. 4b is an uncut, unedited, native gel. Sequencing was performed by GENEWIZ®.

**hiPSC differentiation into day 15 HIO progenitors.** After reaching >95% confluency, cells were differentiated into HIOs using a protocol adapted from refs. [11,13,25]. hiPSC colonies were dissociated into single cells using Gentle Cell Dissociation Reagent (StemCell Technologies cat. no. 07174), and re-plated at a density of $2 \times 10^6$ cells per well of a Matrigel-coated six-well tissue culture plate in mTeSR1 supplemented with Y27632 (Tocris, 5 μM). After 24 h, cells were then differentiated into definitive endoderm using the StemCell Technologies StemDiff Definitive Endoderm Kit (Cat#05110), as per manufacturer's instructions. Cells were then assessed by flow cytometry for anti-CXCR4-PE (ThermoFisher MHCXCR404) and anti-c-kit-APC (Biolegend 323205). At day 3, cells were split 1:3 as described above into new hESC Matrigel® coated 6-well plates, and incubated with DS/SB (see ref. [25]), containing Dorsomorphin (2 μM Stemgent, cat. no. 04-0024) and SB431542 (10 μM Tocris, cat. no. 1614) supplemented with Y27632 for 24 h, followed by DS/SB without Y27632 for 48 h. At day 6, cells were split again 1:3 as described above, and incubated in CB/RA containing CHIR99021 (CHIR) (3 μM, Tocris, cat. no. 4423), rhBMP4 (10 ng/mL, R&D Systems, cat no. 314-BP), and retinoic acid (RA) (100 nM, Sigma, cat. no. R2625-50MG). Basal media for both DS/SB and CBRA consisted of complete serum-free differentiation medium (cSFDM), containing IMDM (ThermoFisher) and Ham's F12 (ThermoFisher) with B27 Supplement with retinoic acid (Invitrogen), N2 Supplement (Invitrogen), 0.1% bovine serum albumin Fraction V (Invitrogen), monothioglycerol (Sigma), Glutamax (ThermoFisher), ascorbic acid (Sigma), and primocin. For a comprehensive list of reagents and catalog numbers, please see Supplementary Table 1, for media recipes, see Supplementary Table 2, and for antibodies, please see Supplementary Table 4.

**Sorting and re-plating of day 15 progenitors into 3D HIOs.** At days 14–15, cells were sorted using the protocol outlined in ref. [25] and the surface marker algorithm described by our group in ref. [11]. Cells were dissociated using 0.05% Trypsin-EDTA (ThermoFisher), and washed in DMEM with 20% FBS. Cells were then strained using a 40 μm filter, spun at $300 \times g$ for 5 min, and resuspended in FACS buffer containing 5 μM Y27632, and stained for CD47-PerCP/Cy5.5 (BioLegend, cat. no. 323110) and CD26-PE (BioLegend, cat. no. 302705) for 30 min on ice, protected from light. Cells were then washed with additional DMEM/20% FBS, spun down again at $300 \times g$ for 5 min, and resuspended in new FACS buffer containing 10 nM Calcein Blue in DMSO (ThermoFisher, cat. no. C1429). Cells were then sorted for either the CD47lo/CD26hi or the CD47lo/GFP+ populations using an operator-assisted MoFlo Astrios EQ (Beckman Coulter) at the Boston University Flow Cytometry Core Facility (FCCF). After sorting, cells were spun down and resuspended in 3D intestinal Matrigel (Corning 354234), in droplets of 50–100 μL (supplemented with media conditions outlined below), at a density of $0.5–1 \times 10^3$ cells/μL, and plated on a pre-warmed 24-well tissue culture plate. After allowing the droplets to solidify for 20 min in a 37 °C incubator, cells were treated with a variety of different media conditions

described in Supplementary Tables 1, 2, supplemented with Y27632. After 3–4 days, fresh media was added without Y27632, and with further media replacement performed every 3–4 days, depending on confluency.

**Passaging of three-dimensional HIOs**. Any media was aspirated, and each well of HIOs was treated with 1 mL of Cell Recovery Solution (Corning cat. no. 354253), and placed at 4 °C for 30 min. Wells were then washed with PBS, and all contents were spun down at $300 \times g$ for 5 min. Organoids were pipetted gently, and resuspended in fresh Matrigel droplets supplemented with media, taking care not to break up organoids. Split densities varied based on original confluency and experimental needs. Organoids were plated and were cultured for up to 100 days with splits every 1–2 weeks as needed.

**Organoid immunofluorescence and microscopy**. Images of whole, live organoids were captured in their tissue culture vessel embedded in 3D Matrigel droplets and submerged in culture media, using a Keyence BZ-X710 All-in-one Fluorescence Microscope. For whole mounts, organoids were dissociated from their Matrigel droplet as described above, washed with PBS, and then fixed in 4% paraformaldehyde (Electron Microscopy Sciences, cat. no. 19208) at room temperature for 30 min. Whole mount HIOs were then washed with PBS, and blocked in 4% normal donkey serum (NDS) with 0.5% Triton X-100 (Sigma) for 30 min. They were then incubated overnight in primary antibody (see Supplementary Table 4) in 0.5% Triton X-100 and 4% NDS. Samples were then washed in 4% NDS and incubated with secondary antibody from Jackson Immunoresearch (1:300 anti-rabbit IgG (H + L), 1:500 anti-chicken IgY, or anti mouse IgG (H + L)) for 45 min at room temperature. Nuclei were stained with Hoechst dye (Thermo Fisher, 1:500). Whole organoids were then mounted with flouromont-G (Southern Biotech) on cavity slides and cover-slipped. For paraffin sectioning, samples were fixed as described above, and washed with PBS. Organoids were then embedded into HistoGel™ Specimen Processing Gel (Richard Allen Scientific), and submitted to the Boston University Experimental Pathology Core Facility for paraffin embedding. Sections were then deparaffinized, followed by an antigen retrieval in a laboratory microwave for 3 min at full power, and 8 min at 30% power, and were set aside to cool for 30 min. Sections were then washed and stained as described above (See Supplementary Table 4 for full list of primary and secondary antibodies). Both stained whole mount and paraffin embedded sections were visualized with either a Zeiss LSM 700 laser scanning confocal microscope or a Nikon Eclipse Ti2 Series Microscope, and processed and analyzed in Fiji.

**Flow cytometry**. Cells for flow cytometry were dissociated using Gentle Cell Dissociation Reagent, followed by resuspension in FACS Buffer comprising of $PBS^{-/-}$ with 0.5% FBS. Antibodies for assessment of definitive endoderm, and the day 15 sort for lung/intestinal progenitors are listed above, with appropriate isotype (IgG1) and unstained controls. To assess EpCam expression, organoids were dissociated as described above, and then further incubated with 0.05% Trypsin for 20 min at 37 °C. After incubation, the reaction was inactivated with DMEM/20% FBS, and the cells were mechanically dissociated by pipetting. Cells were the spun down, resuspended in FACS Buffer, and stained with anti-EpCAM-APC (BioLegend, cat. no. 324208) for 20 min at room temperature, protected from light. Cells were then washed, resuspended in fresh FACS buffer, and strained into BD FACS tubes (Corning cat. no. 352235). All experiments were performed on a BD FACSCalibur™ or STRATEDIGM S1000EON and analyzed using FlowJo.

**Forskolin swelling assay**. Forskolin-induced swelling was performed in organoids at days 29–31 of differentiation, using a similar protocol to previously published work[12,48]. Three independent differentiations were performed for each cell line; organoids were plated in three-dimensional Matrigel (at least six wells per differentiation) and incubated in fresh media for 1–2 days prior to forskolin treatment. Images were taken using a Keyence BZ-X700 fluorescence microscope immediately prior to (time 0 h) and 24 h after (time 24 h) the addition of 5 μM forskolin (Sigma). Imaging analysis was performed using ImageJ; sphere cross-sectional surface area was calculated using a binary analysis of circular (circularity > 0.3) and well sized (area > 900 μm$^2$) organoids. Whole well sphere cross-sectional area at time 0 was set to 1 and the ratio of time 24 h to time 0 cross-sectional area is indicated as normalized cross-sectional area. Time lapse images were captured using a Keyence BZ-X700 microscope with serial imaging of a mapped well (one per condition) every 2.5 min.

**Steady-state lumen area calculation**. As previously developed[51], steady-state lumen area (SLA) was calculated by determining the ratio of lumen to whole organoid cross-sectional area. Using images captured as above, ImageJ was used for quantification (average of 30 organoids per cell line). Epithelial and luminal perimeter was measured manually for each image.

**RNA isolation and qRT-PCR analysis**. RNA was isolated from all samples using the RNeasy Kit (QIAGEN cat. no. 74014), either immediately after dissociation from tissue culture vessels, or after storage at −20 °C in RNAlater (ThermoFisher cat. no. 7020) as per the manufacturer's instructions. RNA was then reverse transcribed to cDNA using the SuperScript™ III First-Strand Synthesis System (Invitrogen cat. no. 18080093) as per the manufacturer's recommended

parameters. RNA was quantified using a NanoDrop™ Lite Spectrophotometer (ThermoFisher) and input was standardized across all samples, to ensure normalized cDNA yields for downstream PCR applications. qRT-PCR was performed using both the TaqMan® or SYBR® Green (Applied Biosystems) master-mixes as per manufacturer's instructions, and the QuantStudio 7 Flex Real-Time 384 Well PCR System with barcoded 384-well plates. Relative fold change above undifferentiated iPSC was determined by calculating the ΔΔCt, using either *GAPDH* (TaqMan) or *ACTB* (SYBR). For primer sequences, see Supplementary Table 3.

**scRNAseq of days 6 and 13 progenitors**. Surrogate wells of differentiated iPSCs (C17) toward endodermal lineages (as described above) from a single MF differentiation were isolated at days 6 and 13. Briefly, cells were disassociated and brought to a single-cell suspension with AccuMax, counted and resuspended in appropriate volume. Cell isolation, capture, and library prep followed the 10x Genomics scRNA-Seq (V2) protocol. Libraries produced were quantified by a Kapp kit and sequenced on an Illumina NextSeq 500. The Cell Ranger software pipeline produced the FASTQ and Counts matrix files. Day 6 library generated 2215 cells at a depth of 53,297 reads/cell with mean genes per cell detected at 4090/cell. Day 13 library generated 2763 cells at a depth of 53,471 with mean genes per cell detected at 3103/cell. Seurat ver. 3.0 was used to further process data. Data was merged and normalized using the regularized negative binomial regression method[76] with cell degradation (i.e., mitochondrial percentage) regressed out during data scaling. Dimensionality reduction methods like PCA and UMAP were used to represent gene expression. Louvain method was used for clustering. Differential expression tests were done with MAST[77]. The dataset supporting the conclusions of this experiment is available in the GEO repository, accession GSE140405.

**Bulk RNA sequencing by digital gene expression**. In order to test differential gene expression, we performed 3′ tag digital gene expression profiling (DGE). Cells (bBU1) were differentiated to day 15, sorted for CD47$^{lo}$ as described above, and plated in 3D Matrigel in CK + DCI and IM + CHIR. At day 42, RNA was isolated from HIOs as described above. In contrast to traditional bulk RNA-Seq, which generates sequencing libraries from the whole transcripts, 3′ tag DGE only covers the terminal fragment of a transcript, complementary to 3′-end sequences[57]. Restricting the sequencing coverage to a small part of the transcripts reduces the number of reads required to profile the full transcriptome. RNA was extracted and amplified from all described samples as described above. Subsequently, library preparation and sequencing were performed at the Broad Institute. Reads were aligned to the ENSEMBL human reference genome GRCh38.9[78] using STAR[79]. We used *edgeR* package[80] to import, filter and normalize the count matrix, followed by the *glimma* package[81] and *voom*[82], for linear modeling and differential expression testing using empirical Bayes moderation to estimate gene-wise variability before significance testing based on the moderated *t*-statistic. We used a corrected *p*-value[83] of 0.05 as threshold to call differentially expressed genes. Functional characterization was done using *Enrichr*[28,29]. The dataset supporting the conclusions of this experiment is available in the GEO repository, accession GSE128922.

**Statistical analysis**. Experimental data for Flow Cytometry and RT-PCR are reported as mean ± s.d. All statistical analysis was performed using GraphPad Prism Software, with statistical significance determined by one way-ANOVA followed by Tukey's test (>2 groups, $n = 3$ per group except IM + CHIR $n = 2$) or student's two-tailed unpaired *T*-test (2 groups, $n = 3$ per group) or paired two-tailed *t*-test, where * = $p < 0.05$, ** = $p < 0.005$, **** = $p < 0.0001$.

**Reporting summary**. Further information on research design is available in the Nature Research Reporting Summary linked to this article.

## Data availability
The authors declare that all data supporting the findings of this study are available within the article and its supplementary material files, or from the corresponding author on reasonable request. The dataset supporting the conclusions of the bulk RNA-sequencing experiment (Fig. 1) has been deposited in the GEO repository under accession code: GSE128922. The dataset supporting the conclusions of the scRNAseq experiment (Fig. 2) has been deposited in the GEO repository under accession code: GSE140405. Further details of iPSC derivation, characterization, and culture are available for free download at http://www.bu.edu/dbin/stemcells/protocols.php.

## Code availability
All data analysis was performed using publicly available methodologies. The bulk RNA sequencing was analyzed using the tools described in the methods section above. The scRNA-seq data was analyzed on a platform using the Cell Ranger and Seurat ver 3.0 pipelines.

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

## Acknowledgements

The authors would like to thank Seonmi Park and Greg Miller for their invaluable technical support. We thank Brian Tilton and the BUMC Flow Cytometry Core for their assistance in cell sorting and flow cytometry experiments, supported by NIH grant 1UL1TR001430, as well as Marianne James and the Boston University iPSC Core for the karyotyping and banking of BU1CG. A.M. is supported by 5T32HL007969-12; A.C., A.J., K.A., and D.H. by TL1TR001410; A.B. by T32HL007035; D.N.K. by R01HL095993 and R01HL128172; F.H. by Cystic Fibrosis Foundation grant HAWKIN15XX0 and NIH grant R01HL139799; and G.M. by R01AI130199 and 1R21NS111499-01. The CReM iPSC Core is supported by NIH 1R24HL123828-01 and U01TR001810.

## Author contributions

A.M. and A.C. performed experimental design, data acquisition, analysis, and interpretation. A.M. wrote the paper. D.H., A.B., A.J., K.A., A.S., M.P., and A.S. performed experiments. M.V. provided technical support. C.V and J.M. performed bioinformatic analyses. D.N.K. and F.H. provided experimental guidance and revised the paper. G.M. designed and supervised experiments, edited, revised, and performed final approval of the paper.

## Competing interests

The authors declare no competing interests.
