## [Peer Review File · Nature Communications]

Reviewers' Comments:

Reviewer #1:

Remarks to the Author:

Overall this is a nice study that presents a defined directed differentiation protocol for the generation of intestinal organoids (and maybe even small intestinal versus colonic organoids) lacking the associated mesenchyme (presumably of mesodermal origin) that is found in the original protocol from the Wells lab. There is clear importance for having a robust, mesenchyme-free human directed differentiation protocol, if for no other reason than to have a reductionist framework for studying the epithelium, and then for potential add-back of different cell types to clearly determine their effects on the epithelium and vice-versa. I have several comments/concerns:

One of my main comments surrounds the lack of mesenchyme that comes along de facto in the original Wells protocol but is absent here. This is a main point of novelty in the current study. Is the lack of mesenchyme in the current MF protocol simply a byproduct of the FACs step? e.g., figure 1b states that vimentin is down in the HIO culture vs. D8, which I'm assuming based on the diagram are pre-sorted gut tube progenitors. Are there mesodermal lineage cells (and cells with mesenchymal identity) in the culture at this D8 timepoint? And are they selected against solely by the FACs step? This should be more thoroughly investigated- ideally in a timecourse comparing the MC and MF protocols as in figure 5, but including more timepoints and more explicit declaration of when sorting takes place relative to analysis. The presence of mesodermal lineage cells early in the two (MC vs. MF) protocols should be looked at closely, both prior to and after any sorting steps. Markers of mesodermal germ layer as well as of mesenchymal cellular identity should be examined. Ultimately, the fact that CDX2+ cells can form organoids without mesenchyme present is, in my opinion, of less value than if the MF protocol can generate epithelial organoids without mesenchyme in the absence of any sorting steps- e.g., that the differentiation strategy itself really selects against mesodermal lineages relative to the MC protocol. This point did not come across clearly in the manuscript.

Another major point is that much (if not all) of the gene expression analysis presented is plotted as fold change above iPSC. This is essentially a meaningless measure as iPSCs should have no (zero) expression of most intestinal-specific genes, and the difference between 20x zero or 400x zero is not meaningful. Gene expression analysis must include a positive control benchmark- ideally human intestinal or colonic organoids generated from biopsy (should be readily available from a variety of sources). Without such a benchmark, the gene expression data is nearly impossible to interpret. Ideally, such data should be presented as expression relative to housekeeping gene, plotting MF organoids w/ various media formulations adjacent to MC organoids and primary colon or intestinal organoids, as appropriate. iPSC levels should also be included as a separate bar in the graphs to serve as a negative control. This is very important in figure 6 where the authors claim different media formulations are driving proximal (SI) vs distal (colon) identity- one must have positive controls to measure against!

Additional points:

-what is the efficiency of organoid formation between the original MC protocol and the newly presented MF protocol? It would be nice to present some low-mag images of organoid cultures originating from the same number of input iPSCs to determine whether the new protocol is more robust than the prior protocol.

-it seems something is missing between the two paragraphs on page 5 – the first paragraph ends with the generation of day 15 progenitors, but how these day 15 progenitors turn to day 42 organoids at the beginning of the second paragraph is unclear. It appears to come later in the paragraph, but the text could use some clarification.

- the authors use the chiron GSK inhibitor and state that it is there to stimulate Wnt signaling. While GSK inhibitors do promote B-catenin stabilization, they also have many, many other effects as GSK has many, many targets beside B-catenin. Can the authors replace the GSK inhibitor with a Wnt ligand and achieve the same effect? This should be a fairly simple experiment which would really nail down the importance of canonical Wnt activity in this process.

-Figure 2- was there a FACS step here as in Fig 1? I'm not exactly sure what this scRNA seq adds, or what the authors conclude from this figure? The authors make statements such as 'the dorsal and ventral foregut endoderm are marked by sox2 and nkx2.1, respectively, while the boundary of sox2 and cdx2 in the developing endoderm marks the foregut from posterior endoderm'. What bearing this has on the study and how this is reflected in the data are unclear. It appears, for example, that there are many sox2 and nkx2.1 double-positive cells here. Also, the statement that Nkx2.1 expression does not overlap w/ cdx2 or sox17 is not immediately clear. Maybe the data should be presented in three colors: Nkx only, cdx (or sox) only, and double positive in a third color? What is the reader meant to conclude from this figure?

-Page 6- the grammar is off here: 'To better define the dynamics of early intestinal specification, we performed single-cell mRNA sequencing (scRNA-seq) to further characterize the progenitor populations during directed differentiation at single cell resolution (Fig. 2a). Was it to 'better define the dynamics of early intestinal specification' or was it to 'further characterize the progenitor populations during directed differentiation'?

-Figure 3b/c- I'm not sure how acceptable showing results from N=2 +/-SEM is when claiming 'significantly more organoids' than in other conditions?

-Figure 4- I'm a bit confused by the genotyping of the targeted iPSCs. The PCR-based validation seems to have a lot of background bands, and the clone used for further study (109) seems to not have a wildtype band? Is this a homozygote? Does the targeting inactivate the CDX allele? It's not clear where the 5' insertion really is? Better maps of the targeted locus should be presented, and ideally a southern blot to verify proper targeting

Reviewer #2:

Remarks to the Author:

This manuscript describes the generation of small intestinal organoids lacking mesenchymal components from hIPS cells using a novel, defined, serum-free directed differentiation method and a CDX2-GFP reporter line. Resulting organoids are shown to be useful for modelling CFTR gene function ex vivo. The fact that the resulting organoids lack any associated mesenchyme is suggested to be a major improvement over existing IPS-based organoid methods for drug screening approaches.

Although the new culture method is novel and potentially useful for both basic research and disease modeling and screening applications, there are a number of important clarifications needed.

Major critique

- Most of the experiments are conducted using a single hIPS line, making it hard to judge general significance/reproducibility of the major findings. Some hIPS lines are known to be differentiation-resistant using other methods, so would be useful to expand the number of lines evaluated.

- I found the title to be somewhat misleading - "regionally patterned" implies selected differentiation into specific regions of the intestine (ie, Duodenum, jejunum, Ileum) - given that

the major finding is that resulting organoids resemble more the small intestine (without evidence of any proximal to distal phenotypic bias) than colon, it would be more accurate to simply state that mesenchyme-free small intestinal organoids are generated.

- Fig 2 – the authors claim the generation of gut progenitors. What about emergence of early intestinal stem cell markers such as TROP2, OLFM4, LGR5? Do these appear at any stage in the differentiation process as would be expected?

There is no discussion of of the SOX9 data in the manuscript

- Fig 3b – in addition to the variations in organoid formation frequency in the different conditions, there are clearly also major differences in the size of the organoids generated – this should be quantified and discussed. Is this due to different proliferation rates, or more indicative of the relative numbers of proliferating progenitors versus differentiated cells present? Are intestinal stem cells expressing known marker genes such as LGR5 present in these organoids as would be predicted?

The title of Fig 3 is a little misleading – do the authors wish to infer that their differentiation method selectively yields proximal small intestine (ie, duodenum), rather than Jejunum or Ilium – or simply small intestine rather than colon? The upregulation of PDX1 might indeed infer specific differentiation towards duodenum, but this is not validated using other marker genes (such as Gip, Ada, GATA4, TRPV6). Should also be noted that PDX1 is upregulated in antral stomach epithelium.

- Fig 4b – why is there apparently still a WT band in the homozygous lane? Difficult to judge specific signals in this gel picture. Has the possibility of non-specific insertion sites been excluded via Southern blot?

- Fig 4i – please show direct co-localization of GFP and CDX2 expression on the organoids to confirm fidelity of the reporter gene expression.

- Fig 5b – it is difficult to see the reported non-epithelial tissue – please provide enhanced images with pointers. In Fig5c, why is there so little CDX2 expression in MC versus MF organoids?

According to CDX2-GFP expression in 5b, there should be very robust CDX2 expression throughout the MC epithelial tissue. Please also include other mesenchyme markers such as alpha-SMA to further confirm the lack of any stromal components.

- Is there any difference in the epithelial phenotype of the MC versus MF protocol (ie, small intestine versus colon)?

- Fig 6 – Please include analyses of intestinal stem cell markers and proliferation markers. The lysozyme staining is not very convincing – are there lysozyme granules visible? Why do the organoids not show evidence of extensive crypt budding as seen in the adult stem cell-derived cultures? Is the lack of Muc2 expression in the IM+CK conditions simply reflecting incomplete differentiation to the goblet cell lineage (and other lineages)? Can this be confirmed using markers of undifferentiated versus differentiated intestinal epithelium? To further confirm this, please include other goblet cell markers such as Muc6 and colon-specific markers such as Muc5B, ATP12A & SCNN1B (see <https://doi.org/10.1016/j.stemcr.2017.10.013>)

- Fig 7b – much better phenotypic characterization of the organoids generated from these CF donor cells is needed. In 5c, the stated upregulation of CFTR expression in the CK+DCI condition does not appear to be significant.

- It is not clear to me why they chose to model CFTR gene function in their hIPS-derived organoids – why would this system better model the disease than the CM method that includes mesenchyme? There is very little novel in the example they chose – it would have been very surprising if the CFTR assay didn't work on these organoids. Indeed, one could argue that since functional interaction of epithelia and mesenchyme are so important in vivo, it would be better to use a system which incorporates both components for disease modelling/drug screening. The authors need to provide better examples and discussion of where a mesenchyme-free system would be beneficial (eg, single cell sequencing to identify novel cell-types or functions?).

Reviewer #3:

Remarks to the Author:

Congratulations on a nice manuscript. I have most expertise in organoid swelling and disease

modeling so will only comment on this in detail.

-The claim that the assay is helpful for disease modeling requires additional data, it currently shows a very minimal set of data. At least a more diverse set of patient samples is needed (ideally multiple CFTR genotypes) and also the effect of CFTR modulator drugs. One would be wanting to demonstrate that different levels of swell readout relate to drug efficacy or clinical phenotype (correlated to published data). I understand that this is not the full message of the ms, but it needs a bit more than this.

-Wt ipsc derived organoids with DmsO swell as well (supplementary movie 2), how do wt organoids on dmsO compare to fsk? Related to this: intestinal organoids from adult stem cells are different between wt and cf donors, prior to fsk stimulation (called the SLA phenotype, check Dekkers et al, Science Transl Med 2016). Presumably this is due to some endogenous cAMP stimulation. This appears also the case in your cultures. Please comment and provide an explanation (are there cAMP raising agents in your culture medium, or factor produced by cells)

-The swell assay kinetics are not fully clear to me: the movies show shrinking and growing organoids, and it requires a long stimulation. Adult stem cell intestinal organoids swell pretty uniformly within 30 min after fsk (wt or CFTR modulator corrected). Why does it take so long? That suggests to me that the cells do not have a lot of CFTR (otherwise they should swell rapidly), or do they have CFTR but no NKCC? Please complement the RNA analysis with CFTR protein data.

-Related to above: how does quantification of swelling work, cross sectional measurements in preselected structures, how does your data look like when you would do simple whole area measurements, and how automated is the current workflow?

Overall: I find it difficult to understand the pro's and cons of this model in relation to other models. Why should one use this model when adult stem cell intestinal organoids can also be grown at large numbers and provide a clinically validated assay (see also Berkers et al, Cell reports 2019). These cells can also be easily obtained via a painless rectal biopsy (a 5 min procedure). I am sure there is value to this model, but find it hard to place as the paper and discussion is very much oriented towards iPSC. A section with a somewhat broader scope in the discussion would bring some additional insight and balance.

Best wishes and good luck with this exciting novel work

Reply to Reviewers

We greatly appreciate the reviewers' time and comments that we have prepared a "nice manuscript" that contains "exciting and novel work", and that "there is clear importance for having a robust, mesenchyme-free human directed differentiation protocol". We have chosen to respond to their comments as follows:

Reviewer #1:

The Absence of Mesenchyme

This is a main point of novelty in the current study. Is the lack of mesenchyme in the current MF protocol simply a byproduct of the FACS step? e.g., figure 1b states that vimentin is down in the HIO culture vs. D8, which I'm assuming based on the diagram are pre-sorted gut tube progenitors. Are there mesodermal lineage cells (and cells with mesenchymal identity) in the culture at this D8 timepoint? And are they selected against solely by the FACS step? This should be more thoroughly investigated- ideally in a time-course comparing the MC and MF protocols as in figure 5, but including more timepoints and more explicit declaration of when sorting takes place relative to analysis. The presence of mesodermal lineage cells early in the two (MC vs. MF) protocols should be looked at closely, both prior to and after any sorting steps. Markers of mesodermal germ layer as well as of mesenchymal cellular identity should be examined. Ultimately, the fact that CDX2+ cells can form organoids without mesenchyme present is, in my opinion, of less value than if the MF protocol can generate epithelial organoids without mesenchyme in the absence of any sorting steps- e.g., that the differentiation strategy itself really selects against mesodermal lineages relative to the MC protocol. This point did not come across clearly in the manuscript.

We appreciate the important questions regarding the dynamics of mesenchymal cell emergence and identity during the MF protocol as compared to the MC protocol. We have included a step by step time-course comparing the early stages of the MC and MF protocols as an addition to Figure 5 that tracks the percentage of EpCam + cells at days 6, 8, and 10 of differentiation (Fig. 5f). We have also analyzed our scRNAseq dataset (that includes cells differentiated using the MC protocol at days 6 and 13 of differentiation) for the expression of EpCAM+ cells at Days 6 and 13 (Fig. 5g), as well as the emergence of mesenchymal and mesodermal markers at these time points, including VIM, COL1A1, COL3A1, FN1, THY1, and ACTA2 which are presented in a new Supplementary Fig. 6. With the exception of VIM and FN1, the vast majority of cells at day 13 did not express the other markers listed above. It has been reported that epithelial cells express mesenchymal genes including VIM and FN1 during early murine organogenesis (E9-11.5) (Dong et al, 2018: PMID 29540203) (Cao et al 2019, 30787437), which may explain why our day 13 cells express these markers, given that they are likely similar to human cells early in development. Furthermore, the vast majority of cells at both days 6 and 13 expressed EpCAM transcript, supporting our flow cytometry data. Additionally, as recommended by the Reviewer, we performed an MF differentiation omitting the FACS step, re-plating all cells at day 15 in 3D Matrigel droplets. We demonstrated that cells at day 30 were ~90% EpCam+, indicating that the MF protocol does in fact select against mesodermal lineages relative to the MC protocol (Supplementary Fig. 6).

Gene Expression Analysis

Another major point is that much (if not all) of the gene expression analysis presented is plotted as fold change above iPSC. This is essentially a meaningless measure as iPSCs should have no (zero) expression of most intestinal-specific genes, and the difference between 20x zero or 400x zero is not meaningful. Gene expression analysis must include a positive control benchmark- ideally human intestinal or colonic organoids generated from biopsy (should be readily available from a variety of sources). Without such a benchmark, the gene expression data is nearly impossible to interpret. Ideally, such data should be presented as expression relative to

housekeeping gene, plotting MF organoids w/ various media formulations adjacent to MC organoids and primary colon or intestinal organoids, as appropriate. iPSC levels should also be included as a separate bar in the graphs to serve as a negative control. This is very important in figure 6 where the authors claim different media formulations are driving proximal (SI) vs distal (colon) identity- one must have positive controls to measure against!

We acknowledge the Reviewer's concerns regarding the plotting of gene expression (qRT-PCR) data as fold change above iPSC, and its physiological relevance to the success or failure of directed differentiation toward a specific lineage. This is a well-known issue in our field and we are sure the Reviewer is aware that a significant number of published studies still use the same approach in the absence of a better alternative (Jacob et al 2017, Hawkins et al 2017, Macauley et al 2017), given that qRT-PCR results are generally only relevant within the context of a given set of experiments. We would like to emphasize that all of the data is calculated as $2^{-\Delta\Delta CT}$, normalized to a housekeeping gene (either GAPDH or β -Actin) and represented as fold change over day zero (undifferentiated iPSCs). Nevertheless, in order to strengthen our analysis and in response to the Reviewer's request, we have now included the addition of an RNA primary control, from adult colon (Fig. 3d, 6d). We hope that the addition of these primary controls will alleviate these concerns and illustrate true biological relevance of our directed differentiation protocol.

Additional Questions

1. What is the efficiency of organoid formation between the original MC protocol and the newly presented MF protocol? It would be nice to present some low-mag images of organoid cultures originating from the same number of input iPSCs to determine whether the new protocol is more robust than the prior protocol.

We would like to highlight the lower magnification images in Figures 3b and 4h, and the quantification in figures 3c and 4g; these were generated from the same number of input hiPSCs (2 million cells/well of 1 six well plate). Additionally, the text has been amended to more clearly illustrate that the yields and images are derived from the same number of input cells. Furthermore, we have added a panel to Supplementary Figure 5b that more clearly demonstrates the dynamics of cellular proliferation during the early stages of differentiation.

2. It seems something is missing between the two paragraphs on page 5 – the first paragraph ends with the generation of day 15 progenitors, but how these day 15 progenitors turn to day 42 organoids at the beginning of the second paragraph is unclear. It appears to come later in the paragraph, but the text could use some clarification.

We apologize for the confusion caused by this unclear language. As detailed in the Methods Section, after sorting and plating the cells on day 15 embedded in 3D Matrigel droplets, the cells are incubated for the following several weeks with media changes every 5 days until analysis. During this period, we observe the emergence of organized 3D epithelial structures and we normally wait until day 40-45 to perform any analysis to allow the intestinal organoids to grow and further specify into epithelial specific cell-types. The language has been amended in the attached revised version of the manuscript.

3. The authors use the chiron GSK inhibitor and state that it is there to stimulate Wnt signaling. While GSK inhibitors do promote B-catenin stabilization, they also have many, many other effects as GSK has many, many targets beside B-catenin. Can the authors replace the GSK inhibitor with a Wnt ligand and achieve the same effect? This should be a fairly simple experiment which would really nail down the importance of canonical Wnt activity in this process.

We appreciate the concern raised over the use of GSK3 inhibitor CHIR99021 as a direct proxy for Wnt signaling activity. Although the original protocol describing the generation of intestinal organoids from hiPSCs (*Spence et al*, 2011) employed Wnt3A as the primary source of Wnt pathway activation, in many subsequent manuscripts describing directed differentiation of intestinal organoids from iPSCs (Munera et al 2017, Kumar et al 2019 30745430, Capeling et al 2019 30612954, McCracken et al 2014 25363776, Watson et al 2015 25326803), CHIR99021 has been used as the primary activator of the Wnt signaling cascade. Additionally, inhibition of GSK-3 β by CHIR has been well characterized as a strong inducer of the Wnt/ β -catenin signaling pathway (Naujok et al, 2014 24779365). The preceding literature has been added to the discussion to emphasize the widespread usage of CHIR 99021 as a Wnt/ β -catenin agonist. While it is still possible that CHIR may have additional effects independent of Wnt stimulation, that is beyond the scope of these studies.

4. Figure 2- was there a FACS step here as in Fig 1? I'm not exactly sure what this scRNA seq adds, or what the authors conclude from this figure? The authors make statements such as 'the dorsal and ventral foregut endoderm are marked by sox2 and nkx2.1, respectively, while the boundary of sox2 and cdx2 in the developing endoderm marks the foregut from posterior endoderm'. What bearing this has on the study and how this is reflected in the data are unclear. It appears, for example, that there are many sox2 and nkx2.1 double-positive cells here. Also, the statement that Nkx2.1 expression does not overlap w/ cdx2 or sox17 is not immediately clear. Maybe the data should be presented in three colors: Nkx only, cdx (or sox) only, and double positive in a third color? What is the reader meant to conclude from this figure?

The use of dual SMAD inhibition was originally reported to strongly induce anterior foregut specification (Green et al). In order to further investigate our findings that dual SMAD inhibition did not preclude differentiation into posterior endodermal lineages, we performed this scRNA-seq experiment to demonstrate this at single cell resolution. In order to more clearly visualize the data, Figure 2 has been amended significantly, along with the accompanying text. The data has been re-analyzed and presented using a novel pipeline, further supporting our previous claim that the day 6 and day 13 cells do, in fact, cluster independently in an unsupervised manner. We thank the Reviewer for the suggestion on how best to present the data. The text has also been amended to clarify the experimental design and to more accurately reflect the experimental schematic presented in Fig. 2a.

5. Page 6- the grammar is off here: 'To better define the dynamics of early intestinal specification, we performed single-cell mRNA sequencing (scRNA-seq) to further characterize the progenitor populations during directed differentiation at single cell resolution (Fig. 2a). Was it to 'better define the dynamics of early intestinal specification' or was it to 'further characterize the progenitor populations during directed differentiation'?

We have amended the language to more clearly illustrate our goal of characterizing the progenitor populations during directed differentiation. See previous response.

6. Figure 3b/c- I'm not sure how acceptable showing results from N=2 +/-SEM is when claiming 'significantly more organoids' than in other conditions?

While we acknowledge that sample size used in that experiment was limited, the large number of media conditions screened made it highly impractical and challenging to have a larger N. The experiment was nevertheless informative as we were able to confirm the same result using a

different hiPSC line in an independent experiment (Fig. 4f). The language has been amended to more accurately depict the data shown in Figure 3b/c.

7. Figure 4- I'm a bit confused by the genotyping of the targeted iPSCs. The PCR-based validation seems to have a lot of background bands, and the clone used for further study (109) seems to not have a wildtype band? Is this a homozygote? Does the targeting inactivate the CDX allele? It's not clear where the 5' insertion really is? Better maps of the targeted locus should be presented, and ideally a southern blot to verify proper targeting

We appreciate your comments regarding the characterization of our CDX2-eGFP iPSC reporter line. We have re-performed the PCR screening in order to more clearly illustrate that the clone used in our experiments (109) does in fact contain a bi-allelic insertion of our eGFP donor construct and therefore there is no wild-type band in the PCR. We have also included sequencing of the region of interest demonstrating a bi-allelic insertion, as well as sequencing of the three most likely off target sites as determined by the sequence of the chosen sgRNA (NHLRC4, RAI4, and SPP3) (demonstrating no insertion) as an addition to Supplementary Figure 4 (new Supplementary Fig. 4c). We have also amended Fig. 4a to include a clearer map of the targeted locus, and have added language in the manuscript to clarify that, due to the design of our donor construct and insertion sites, CDX2 is not, in fact, inactivated as a result of gene editing as the GFP is linked to the full CDX2 cDNA via a 2A peptide sequence.

Reviewer #2:

Major Critiques

1. Most of the experiments are conducted using a single hiPSC line, making it hard to judge general significance/reproducibility of the major findings. Some hiPSC lines are known to be differentiation-resistant using other methods, so would be useful to expand the number of lines evaluated.

We apologize for the lack of clarity on the number of lines throughout the manuscript. Figure legends and text have been updated to more clearly demonstrate which lines are used for each experiment. We appreciate that iPSCs can display significant line to line variability in directed differentiation. However, throughout our studies we used five independent hiPSC lines derived from four separate genetic backgrounds (BU3NG, BU1CG, C17, RC204, and RC204-corr) across multiple independent experiments. Overall the data from all four lines showed the robustness and consistency of our new protocol.

2. I found the title to be somewhat misleading – “regionally patterned” implies selected differentiation into specific regions of the intestine (ie, Duodenum, jejunum, Ileum) – given that the major finding is that resulting organoids resemble more the small intestine (without evidence of any proximal to distal phenotypic bias) than colon, it would be more accurate to simply state that mesenchyme-free small intestinal organoids are generated.

Our decision to include in the title the terms “regionally patterned” reflects our findings that depending on the specific conditions used for differentiation the HIOs showed a more proximal (small intestine) vs distal (colonic) identity. Nevertheless, we acknowledge and welcome the Reviewer comments and we decided to change the title accordingly.

3. Fig 2 – the authors claim the generation of gut progenitors. What about emergence of early intestinal stem cell markers such as TROP2, OLFM4, LGR5? Do these appear at any stage in the differentiation process as would be expected? There is no discussion of the SOX9 data in the manuscript

In order to address these questions, we have reanalyzed the scRNA-seq data. We have now added a new panel to Figure 2 (Fig. 2d), that demonstrates that there are a significant number of cells at day 6 that express TROP2, while a subset of cells at day 13 express LGR5. OLFM4 transcripts were detected in a small subset of cells at day 6 and were not detected at day 13. Additionally, we were pleased to see that relatively few cells expressed SOX9 at day 13, suggesting that the majority of our CDX2+ cells at day 13 were likely giving rise to intestine instead of other lineages such as pancreas. This was added to the text of the results section.

4. Fig 3b – in addition to the variations in organoid formation frequency in the different conditions, there are clearly also major differences in the size of the organoids generated – this should be quantified and discussed. Is this due to different proliferation rates, or more indicative of the relative numbers of proliferating progenitors versus differentiated cells present? Are intestinal stem cells expressing known marker genes such as LGR5 present in these organoids as would be predicted?

We appreciate the insightful observation of variance in organoid size in addition to proliferation rate. Organoid size differences (diameter) between IM+CK and CK+DCI organoids were quantified; while CK+DCI HIOs were, on average, larger, the difference was not statistically significant (data not shown). We have added a sentence in the Discussion Section to reflect this. Regarding expression of stem cell markers please see previous answer.

5. The title of Fig 3 is a little misleading – do the authors wish to infer that their differentiation method selectively yields proximal small intestine (ie, duodenum), rather than Jejunum or Ilium – or simply small intestine rather than colon? The upregulation of PDX1 might indeed infer specific differentiation towards duodenum, but this is not validated using other marker genes (such as Gip, Ada, GATA4, TRPV6). Should also be noted that PDX1 is upregulated in antral stomach epithelium.

It is well accepted in the field that simultaneous expression of CDX2 and PDX1 demarks duodenal identity (Spence et al, 2011; McCracken et al 2011; Watson et al 2014). In order to more specifically characterize our proximal intestinal organoids, and as suggested by the Reviewer, we have now assessed transcription of GATA4 by qRT-PCR, confirming the duodenal identity of the IM-CK generated HIOs. This new data has been added to Figures 3d and 6d.

6. Fig 4b – why is there apparently still a WT band in the homozygous lane? Difficult to judge specific signals in this gel picture. Has the possibility of non-specific insertion sites been excluded via Southern blot?

Please see answer to Reviewer 1 question 7.

7. Fig 4i – please show direct co-localization of GFP and CDX2 expression on the organoids to confirm fidelity of the reporter gene expression.

Please see the revised Fig. 4f, where Day 40 HIOs were stained for GFP and CDX2, depicting nuclear CDX2 and pan-cellular GFP staining in CDX2 positive cells, while CDX2 negative cells are also negative for GFP, confirming the fidelity of the reporter gene expression.

8. Fig 5b – it is difficult to see the reported non-epithelial tissue – please provide enhanced images with pointers. In Fig5c, why is there so little CDX2 expression in MC versus MF organoids? According to CDX2-GFP expression in 5b, there should be very robust CDX2 expression

throughout the MC epithelial tissue. Please also include other mesenchyme markers such as alpha-SMA to further confirm the lack of any stromal components.

We have updated the images to include arrowheads (Fig. 5b) demarking areas with non-epithelial outgrowth, that are clearly GFP-. Fig. 5c shows robust CDX2 staining in the MC-derived HIOs, although there are a significant number of CDX2 negative Vimentin positive areas, as a result of the mesenchymal component in MC-derived organoids.

9. Is there any difference in the epithelial phenotype of the MC versus MF protocol (ie, small intestine versus colon)?

We did not observe any noticeable difference in the epithelial phenotype of the MC vs MF protocol. As briefly stated in the introduction to this letter, it is not our intention to make this manuscript a side by side comparison of our new protocol to previously published protocols. Importantly, we did not intend to state that our protocol is better. The main message is to provide the scientific community with a novel protocol that takes a different developmental path to robustly generate intestinal organoids in the absence of mesenchymal support.

10. Fig 6 – Please include analyses of intestinal stem cell markers and proliferation markers. The lysozyme staining is not very convincing – are there lysozyme granules visible? Why do the organoids not show evidence of extensive crypt budding as seen in the adult stem cell-derived cultures? Is the lack of Muc2 expression in the IM+CK conditions simply reflecting incomplete differentiation to the goblet cell lineage (and other lineages)? Can this be confirmed using markers of undifferentiated versus differentiated intestinal epithelium? To further confirm this, please include other goblet cell markers such as Muc6 and colon-specific markers such as Muc5B, ATP12A & SCNN1B(see <https://doi.org/10.1016/j.stemcr.2017.10.013>)

See response to question 3. We decided to show a low magnification of the lysozyme staining in order to emphasize the correct ratio of lysozyme positive cells within the HIOs, as shown in other HIOs reports. While formation of crypt budding is variable in iPSC-derived HIOs, we do frequently find epithelial organoids that show complex structures (Fig. 3b and e, Fig. 4h and i). We acknowledge that crypt-derived HIOs are likely different from iPSC-derived HIOs, an issue that deserves further investigation which is beyond the scope of this manuscript. MUC2 is expressed mainly in the colon and therefore we do not expect to see much *Muc2* staining in the IM-CK conditions that promote proximal specification.

11. Fig 7b – much better phenotypic characterization of the organoids generated from these CF donor cells is needed. In 5c, the stated upregulation of CFTR expression in the CK+DCI condition does not appear to be significant.

The reviewer raises an interesting and important point. This model provides an opportunity to study the consequences of a CFTR mutation in intestinal epithelium while controlling for genetic background. Are there phenotypic differences as a result of molecular and or/patterning consequences of lacking functional CFTR? We plan on pursuing these important questions in a future manuscript. In Fig. 7 we focused on demonstrating that the HIOs obtained in our protocol were functionally capable of responding to CFTR stimulation, as should be expected in physiologically relevant intestinal epithelium. In the main text we stated that in both conditions CFTR expression was higher than undifferentiated day 0 iPSCs. We have also added a primary colon control to the transcriptional analysis.

12. *It is not clear to me why they chose to model CFTR gene function in their hIPS-derived organoids – why would this system better model the disease than the CM method that includes mesenchyme? There is very little novel in the example they chose – it would have been very surprising if the CFTR assay didn't work on these organoids. Indeed, one could argue that since functional interaction of epithelia and mesenchyme are so important in vivo, it would be better to use a system which incorporates both components for disease modelling/drug screening. The authors need to provide better examples and discussion of where a mesenchyme-free system would be beneficial (eg, single cell sequencing to identify novel cell-types or functions?).*

The functional interaction of intestinal epithelium with intestinal mesenchymal lineages is certainly important and of interest. However, for certain questions the ability to study purified epithelial-only cells is an advantage. We specifically selected CF because within the intestine it is primarily expressed in the epithelium and any effects on the mesenchyme are presumed secondary to the epithelial defect. An epithelial only model facilitates the simple measurement of the level of CFTR function in a reductionist, mesenchyme free manner with the future goal of identifying drug rescue. We have updated the text as recommended to explain our rationale for selecting CF. See also comments to Reviewer 3.

Reviewer #3:

1. *The claim that the assay is helpful for disease modeling requires additional data, it currently shows a very minimal set of data. At least a more diverse set of patient samples is needed (ideally multiple CFTR genotypes) and also the effect of CFTR modulator drugs. One would be wanting to demonstrate that different levels of swell readout relate to drug efficacy or clinical phenotype (correlated to published data). I understand that this is not the full message of the ms, but it needs a bit more than this.*

We agree that to better understand the clinical potential of this platform a broader dataset is required. However, the main theme of this manuscript is the development of a new intestinal differentiation protocol that does not rely on the presence of mesoderm. The data shown in Figure 7 are meant to provide a proof-of-concept of the potential functional relevance of our HIOs, using as an example the measure of CFTR function, which we believe was demonstrated in the dataset presented in Figure 7. We focused on the CF vs CF corrected, genetically matched pair to specifically address in this manuscript if the phenotype was purely CFTR mediated. Based on the comprehensive literature on rectal organoids (Dekkers et al., Nature 2014 and Science Translational Medicine 2016) a full characterization of the robustness of iPSC-derived intestinal organoids to detect drug responsiveness, as the reviewer points out, will require many WT and CF genotypes. These efforts are underway and will be the major focus of a separate manuscript dedicated to these questions.

2. *Wt ipsc derived organoids with DmsO swell as well (supplementary movie 2), how do wt organoids on dmsO compare to fsk? Related to this: intestinal organoids from adult stem cells are different between wt and cf donors, prior to fsk stimulation (called the SLA phenotype, check Dekkers et al, Science Transl Med 2016). Presumably this is due to some endogenous cAMP stimulation. This appears also the case in your cultures. Please comment and provide an explanation (are there cAMP raising agents in your culture medium, or factor produced by cells)*

Thank you for the comments. We have added an updated schematic to describe the workflow, updated and new analyses to address the reviewer's questions and clarify the methodologies and results (see updated Fig. 7a,c,d,e,g) and supplemental figure 7a and b). We are very familiar with the seminal work of Dr. Dekkers and Beekman in developing the rectal organoid assay for CF.

Firstly, there is cAMP in our baseline media and similar to rectal organoid literature (Dekker, Science Transl Med 2016) we observe baseline swelling in the WT organoids. The added cAMP causes swelling in similar magnitude to forskolin alone (data not shown). cAMP and IBMX are part of the directed differentiation protocol media. Based on comments from the reviewer we have included additional analyses based on rectal organoid assays including (1) baseline organoid size and (2) SLA in WT vs CF organoids prior to forskolin stimulation. These data confirm that the differences in size and SLA are CFTR-dependent (Fig. 7d-e). In response to DMSO (vs Fsk), delF508 organoids do not swell significantly so this should not affect future drug screening experiments. However, WT organoids do swell in response to DMSO control but it should be noted that in the experiments shown organoids also receive fresh cAMP in the base media. Future work will be directed at stream-lining the swelling protocol and determining whether cAMP can be removed from the media at later timepoints to remove the baseline swelling effect.

3. The swell assay kinetics are not fully clear to me: the movies show shrinking and growing organoids, and it requires a long stimulation. Adult stem cell intestinal organoids swell pretty uniformly within 30 min after fsk (wt or CFTR modulator corrected). Why does it take so long? That suggests to me that the cells do not have a lot of CFTR (otherwise they should swell rapidly), or do they have CFTR but no NKCC? Please complement the RNA analysis with CFTR protein data.

Thank you for the observation. We have added supplementary material (Supplementary Fig. 7a) highlighting the kinetics of the fsk-induced swelling assay over time in WT HIOs, in order to fully clarify the kinetics of the assay. HIOs actually do begin to swell quite early (30 mins) after Fsk treatment, but at a slower rate than published data of adult stem cell-derived intestinal organoids. We reasoned that this was simply due to lower levels of CFTR mRNA and protein. In response to the reviewer's comments we also added a primary colon control to the qRT-PCR analysis of CFTR expression which confirmed increased expression of CFTR in a primary adult compared to our HIOs.

4. Related to above: how does quantification of swelling work, cross sectional measurements in preselected structures, how does your data look like when you would do simple whole area measurements, and how automated is the current workflow?

We have updated the methods section to provide more information on the analysis and work flow. In addition we have included SLA and baseline organoid size to substantiate our findings. Currently, the work flow is not automated. In future work we hope to assess the feasibility of scaling up the work flow.

5. Overall: I find it difficult to understand the pro's and cons of this model in relation to other models. Why should one use this model when adult stem cell intestinal organoids can also be grown at large numbers and provide a clinically validated assay (see also Berkers et al, Cell reports 2019). These cells can also be easily obtained via a painless rectal biopsy (a 5 min procedure). I am sure there is value to this model, but find it hard to place as the paper and discussion is very much oriented towards iPSC. A section with a somewhat broader scope in the discussion would bring some additional insight and balance.

The reviewer raises a helpful perspective on our discussion. We would like to emphasize that it was not our intention to claim that the data presented in Figure 7 represents a better model than other existing models, including adult stem cell derived HIOs. There are many cell-based models in various stages of characterization and development (Clancy et al, Journal of Cystic Fibrosis, 2019). Primary bronchial epithelial cells, rectal organoids and nasal epithelial cells are amongst

the better characterized. The potential of the iPSC system remains to be fully determined and will require careful study compared to existing platforms. However, there are some unique aspects of this platform including the ability to perform gene-correction in iPSCs and the potential for large scale expansion that could address specific questions in CF, including: 1) a platform to study the role of genetic modifiers and better understand the dramatic heterogeneity seen amongst patients with the same mutation 2) increase the supply of relevant cell types from patients with non-sense CFTR mutations that are now the major focus of the field. A new section has been added to the discussion to provide a broader overview of cell-based platforms for CF and elaborate the potential benefits and limitations of using iPSC-derived HIOs.

Reviewers' Comments:

Reviewer #1:

Remarks to the Author:

My comments/concerns have been well-addressed, I think the paper is in much better shape and fine for publication. One thing that may be good to add with respect to figure 2 (scRNAseq)- the authors look at intestinal markers like *Olfm4/Lgr5*, but see minimal expression. This may not be surprising as these cells are likely still in a fetal state. Thus, it would be great if the authors could pop in a few UMAPs of transcripts representing the fetal ISC signature (<https://www.ncbi.nlm.nih.gov/pmc/articles/PMC6042247/> and <https://www.ncbi.nlm.nih.gov/pmc/articles/PMC3858813/>).

Reviewer #2:

Remarks to the Author:

Major critique satisfactorily addressed

Reviewer #3:

Remarks to the Author:

Thanks and job well done

Response Letter to Reviewer Comments

We would like to thank the reviewers for their time, effort, and thoughtful comments and suggestions, which we feel have greatly strengthened our manuscript. We have chosen to respond to their final comments as follows:

Reviewer #1 (Remarks to the Author):

My comments/concerns have been well-addressed, I think the paper is in much better shape and fine for publication. One thing that may be good to add with respect to figure 2 (scRNAseq)- the authors look at intestinal markers like Olfm4/Lgr5, but see minimal expression. This may not be surprising as these cells are likely still in a fetal state. Thus, it would be great if the authors could pop in a few UMAPs of transcripts representing the fetal ISC signature
(<https://www.ncbi.nlm.nih.gov/pmc/articles/PMC6042247/> and <https://www.ncbi.nlm.nih.gov/pmc/articles/PMC3858813/>).

We thank the Reviewer for raising this important issue. While the expression of fetal markers is of potential interest, we feel that adding new data showing expression of fetal markers in our single cell panel may confuse readers as to the ultimate message of the manuscript, particularly given that the single cell data focuses on day 6 and 13 of differentiation, which is actually very early during our MF directed differentiation protocol. As a matter of fact, it would not be surprising to see expression of fetal markers that early in the differentiation. Importantly, LGR5 has yet to be proven as a definitive marker of intestinal stem cells in humans, and therefore it is hard to fully interpret the significance of the low expression shown in day 6 and 13.

Reviewer #2 (Remarks to the Author):

Major critique satisfactorily addressed

We thank the reviewer for their time, and are pleased that we were able to address their critiques.

Reviewer #3 (Remarks to the Author):

Thanks and job well done

We thank the reviewer for their time, and are pleased that we were able to address their critiques.